# Copeland Dueling Bandits

**Masrour Zoghi**
Informatics Institute
University of Amsterdam, Netherlands
m.zoghi@uva.nl

**Zohar Karnin**
Yahoo Labs
New York, NY
zkarnin@yahoo-inc.com

**Shimon Whiteson**
Department of Computer Science
University of Oxford, UK
shimon.whiteson@cs.ox.ac.uk

**Maarten de Rijke**
Informatics Institute
University of Amsterdam
derijke@uva.nl

## Abstract

A version of the dueling bandit problem is addressed in which a *Condorcet winner* may not exist. Two algorithms are proposed that instead seek to minimize regret with respect to the *Copeland winner*, which, unlike the Condorcet winner, is guaranteed to exist. The first, **Copeland Confidence Bound (CCB)**, is designed for small numbers of arms, while the second, **Scalable Copeland Bandits (SCB)**, works better for large-scale problems. We provide theoretical results bounding the regret accumulated by CCB and SCB, both substantially improving existing results. Such existing results either offer bounds of the form $\mathcal{O}(K \log T)$ but require restrictive assumptions, or offer bounds of the form $\mathcal{O}(K^2 \log T)$ without requiring such assumptions. Our results offer the best of both worlds: $\mathcal{O}(K \log T)$ bounds without restrictive assumptions.

## 1 Introduction

The *dueling bandit problem* [1] arises naturally in domains where feedback is more reliable when given as a pairwise preference (e.g., when it is provided by a human) and specifying real-valued feedback instead would be arbitrary or inefficient. Examples include *ranker evaluation* [2, 3, 4] in information retrieval, ad placement and recommender systems. As with other *preference learning* problems [5], feedback consists of a pairwise preference between a selected pair of arms, instead of scalar reward for a single selected arm, as in the $K$-armed bandit problem.

Most existing algorithms for the dueling bandit problem require the existence of a Condorcet winner, which is an arm that beats every other arm with probability greater than $0.5$. If such algorithms are applied when no Condorcet winner exists, no decision may be reached even after many comparisons. This is a key weakness limiting their practical applicability. For example, in industrial ranker evaluation [6], when many rankers must be compared, each comparison corresponds to a costly live experiment and thus the potential for failure if no Condorcet winner exists is unacceptable [7].

This risk is not merely theoretical. On the contrary, recent experiments on $K$-armed dueling bandit problems based on information retrieval datasets show that dueling bandit problems without Condorcet winners arise regularly in practice [8, Figure 1]. In addition, we show in Appendix C.1 in the supplementary material that there are realistic situations in ranker evaluation in information retrieval in which the probability that the Condorcet assumption holds, decreases rapidly as the number of arms grows. Since the $K$-armed dueling bandit methods mentioned above do not provide regret bounds in the absence of a Condorcet winner, applying them remains risky in practice. Indeed, we demonstrate empirically the danger of applying such algorithms to dueling bandit problems that do not have a Condorcet winner (cf. Appendix A in the supplementary material).

The non-existence of the Condorcet winner has been investigated extensively in social choice theory, where numerous definitions have been proposed, without a clear contender for the most suitable resolution [9]. In the dueling bandit context, a few methods have been proposed to address this issue, e.g., SAVAGE [10], PBR [11] and RankEl [12], which use some of the notions proposed by

social choice theorists, such as the Copeland score or the Borda score to measure the quality of each arm, hence determining what constitutes the best arm (or more generally the top-$k$ arms). In this paper, we focus on finding Copeland winners, which are arms that beat the greatest number of other arms, because it is a natural, conceptually simple extension of the Condorcet winner.

Unfortunately, the methods mentioned above come with bounds of the form $\mathcal{O}(K^2 \log T)$. In this paper, we propose two new $K$-armed dueling bandit algorithms for the Copeland setting with significantly improved bounds.

The first algorithm, called **Copeland Confidence Bound (CCB)**, is inspired by the recently proposed Relative Upper Confidence Bound method [13], but modified and extended to address the unique challenges that arise when no Condorcet winner exists. We prove anytime high-probability and expected regret bounds for CCB of the form $\mathcal{O}(K^2 + K \log T)$. Furthermore, the denominator of this result has much better dependence on the "gaps" arising from the dueling bandit problem than most existing results (cf. Sections 3 and 5.1 for the details).

However, a remaining weakness of CCB is the additive $\mathcal{O}(K^2)$ term in its regret bounds. In applications with large $K$, this term can dominate for any experiment of reasonable duration. For example, at Bing, 200 experiments are run concurrently on any given day [14], in which case the duration of the experiment needs to be longer than the age of the universe in nanoseconds before $K \log T$ becomes significant in comparison to $K^2$.

Our second algorithm, called **Scalable Copeland Bandits (SCB)**, addresses this weakness by eliminating the $\mathcal{O}(K^2)$ term, achieving an expected regret bound of the form $\mathcal{O}(K \log K \log T)$. The price of SCB's tighter regret bounds is that, when two suboptimal arms are close to evenly matched, it may waste comparisons trying to determine which one wins in expectation. By contrast, CCB can identify that this determination is unnecessary, yielding better performance unless there are very many arms. CCB and SCB are thus complementary algorithms for finding Copeland winners.

Our main contributions are as follows:

1. We propose two algorithms that address the dueling bandit problem in the absence of a Condorcet winner, one designed for problems with small numbers of arms and the other scaling well with the number of arms.
2. We provide regret bounds that bridge the gap between two groups of results: those of the form $\mathcal{O}(K \log T)$ that make the Condorcet assumption, and those of the form $\mathcal{O}(K^2 \log T)$ that do not make the Condorcet assumption. Our bounds are similar to those of the former but are as broadly applicable as the latter. Furthermore, the result for CCB has substantially better dependence on the gaps than the second group of results.
3. We include an empirical evaluation of CCB and SCB using a real-life problem arising from information retrieval (IR). The experimental results mirror the theoretical ones.

## 2   Problem Setting

Let $K \geq 2$. The $K$-*armed dueling bandit* problem [1] is a modification of the $K$-*armed bandit* problem [15]. The latter considers $K$ arms $\{a_1, \ldots, a_K\}$ and at each *time-step*, an arm $a_i$ can be *pulled*, generating a *reward* drawn from an unknown stationary distribution with expected value $\mu_i$. The $K$-armed *dueling* bandit problem is a variation in which, instead of pulling a single arm, we choose a pair $(a_i, a_j)$ and receive one of them as the better choice, with the probability of $a_i$ being picked equal to an unknown constant $p_{ij}$ and that of $a_j$ being picked equal to $p_{ji} = 1 - p_{ij}$. A problem instance is fully specified by a *preference matrix* $\mathbf{P} = [p_{ij}]$, whose $ij$ entry is equal to $p_{ij}$.

Most previous work assumes the existence of a *Condorcet winner* [10]: an arm, which without loss of generality we label $a_1$, such that $p_{1i} > \frac{1}{2}$ for all $i > 1$. In such work, regret is defined relative to the Condorcet winner. However, Condorcet winners do not always exist [8, 13]. In this paper, we consider a formulation of the problem that does not assume the existence of a Condorcet winner.

Instead, we consider the *Copeland dueling bandit problem*, which defines regret with respect to a *Copeland winner*, which is an arm with maximal *Copeland score*. The Copeland score of $a_i$, denoted Cpld($a_i$), is the number of arms $a_j$ for which $p_{ij} > 0.5$. The *normalized Copeland score*, denoted cpld($a_i$), is simply $\frac{\text{Cpld}(a_i)}{K-1}$. Without loss of generality, we assume that $a_1, \ldots, a_C$ are the Copeland winners, where $C$ is the number of Copeland winners. We define regret as follows:

**Definition 1.** The **regret** incurred by comparing $a_i$ and $a_j$ is $2\text{cpld}(a_1) - \text{cpld}(a_i) - \text{cpld}(a_j)$.

**Remark 2.** Since our results (see §5) establish bounds on the number of queries to non-Copeland winners, they can also be applied to other notions of regret.

## 3 Related Work

Numerous methods have been proposed for the $K$-armed dueling bandit problem, including Interleaved Filter [1], Beat the Mean [3], Relative Confidence Sampling [8], Relative Upper Confidence Bound (RUCB) [13], Doubler and MultiSBM [16], and mergeRUCB [17], all of which require the existence of a Condorcet winner, and often come with bounds of the form $\mathcal{O}(K \log T)$. However, as observed in [13] and Appendix C.1, real-world problems do not always have Condorcet winners.

There is another group of algorithms that do not assume the existence of a Condorcet winner, but have bounds of the form $\mathcal{O}(K^2 \log T)$ in the Copeland setting: Sensitivity Analysis of VAriables for Generic Exploration (SAVAGE) [10], Preference-Based Racing (PBR) [11] and Rank Elicitation (RankEl) [12]. All three of these algorithms are designed to solve more general or more difficult problems, and they solve the Copeland dueling bandit problem as a special case.

This work bridges the gap between these two groups by providing algorithms that are as broadly applicable as the second group but have regret bounds comparable to those of the first group. Furthermore, in the case of the results for CCB, rather than depending on the smallest gap between arms $a_i$ and $a_j$, $\Delta_{\min} := \min_{i>j} |p_{ij} - 0.5|$, as in the case of many results in the Copeland setting,[1] our regret bounds depend on a larger quantity that results in a substantially lower upper-bound, cf. §5.1.

In addition to the above, bounds have been proven for other notions of winners, including Borda [10, 11, 12], Random Walk [11, 18], and very recently von Neumann [19]. The dichotomy discussed also persists in the case of these results, which either rely on restrictive assumptions to obtain a linear dependence on $K$ or are more broadly applicable, at the expense of a quadratic dependence on $K$. A natural question for future work is whether the improvements achieved in this paper in the case of the Copeland winner can be obtained in the case of these other notions as well. We refer the interested reader to Appendix C.2 for a numerical comparison of these notions of winners in practice. More generally, there is a proliferation of notions of winners that the field of Social Choice Theory has put forth and even though each definition has its merits, it is difficult to argue for any single definition to be superior to all others.

A related setting is that of *partial monitoring games* [20]. While a dueling bandit problem can be modeled as a partial monitoring problem, doing so yields weaker results. In [21], the authors present problem-dependent bounds from which a regret bound of the form $\mathcal{O}(K^2 \log T)$ can be deduced for the dueling bandit problem, whereas our work achieves a linear dependence in $K$.

## 4 Method

We now present two algorithms that find Copeland winners.

### 4.1 Copeland Confidence Bound (CCB)

CCB (see Algorithm 1) is based on the principle of *optimism followed by pessimism*: it maintains optimistic and pessimistic estimates of the preference matrix, i.e., matrices $\mathbf{U}$ and $\mathbf{L}$ (Line 6). It uses $\mathbf{U}$ to choose an *optimistic Copeland winner* $a_c$ (Lines 7–9 and 11–12), i.e., an arm that has some chance of being a Copeland winner. Then, it uses $\mathbf{L}$ to choose an *opponent* $a_d$ (Line 13), i.e., an arm deemed likely to discredit the hypothesis that $a_c$ is indeed a Copeland winner.

More precisely, an optimistic estimate of the Copeland score of each arm $a_i$ is calculated using $\mathbf{U}$ (Line 7), and $a_c$ is selected from the set of top scorers, with preference given to those in a shortlist, $\mathcal{B}_t$ (Line 11). These are arms that have, roughly speaking, been optimistic winners throughout history. To maintain $\mathcal{B}_t$, as soon as CCB discovers that the optimistic Copeland score of an arm is lower than the pessimistic Copeland score of another arm, it purges the former from $\mathcal{B}_t$ (Line 9B).

The mechanism for choosing the opponent $a_d$ is as follows. The matrices $\mathbf{U}$ and $\mathbf{L}$ define a confidence interval around $p_{ij}$ for each $i$ and $j$. In relation to $a_c$, there are three types of arms: (1) arms $a_j$ s.t. the confidence region of $p_{cj}$ is strictly above 0.5, (2) arms $a_j$ s.t. the confidence region of $p_{cj}$ is strictly below 0.5, and (3) arms $a_j$ s.t. the confidence region of $p_{cj}$ contains 0.5. Note that an arm of type (1) or (2) at time $t'$ may become an arm of type (3) at time $t > t'$ even without queries to the corresponding pair as the size of the confidence intervals increases as time goes on.

**Algorithm 1** Copeland Confidence Bound

---

**Input:** A Copeland dueling bandit problem and an exploration parameter $\alpha > \frac{1}{2}$.
1: $\mathbf{W} = [w_{ij}] \leftarrow \mathbf{0}_{K \times K}$ // 2D array of wins: $w_{ij}$ is the number of times $a_i$ beat $a_j$
2: $\mathcal{B}_1 = \{a_1, \ldots, a_K\}$ // potential best arms
3: $\mathcal{B}_1^i = \varnothing$ for each $i = 1, \ldots, K$ // potential to beat $a_i$
4: $\overline{L}_C = K$ // estimated max losses of a Copeland winner
5: **for** $t = 1, 2, \ldots$ **do**
6:     $\mathbf{U} := [u_{ij}] = \frac{\mathbf{W}}{\mathbf{W} + \mathbf{W}^T} + \sqrt{\frac{\alpha \ln t}{\mathbf{W} + \mathbf{W}^T}}$ and $\mathbf{L} := [l_{ij}] = \frac{\mathbf{W}}{\mathbf{W} + \mathbf{W}^T} - \sqrt{\frac{\alpha \ln t}{\mathbf{W} + \mathbf{W}^T}}$, with $u_{ii} = l_{ii} = \frac{1}{2}, \forall i$
7:     $\overline{\text{Cpld}}(a_i) = \# \left\{ k \mid u_{ik} \geq \frac{1}{2}, k \neq i \right\}$ and $\underline{\text{Cpld}}(a_i) = \# \left\{ k \mid l_{ik} \geq \frac{1}{2}, k \neq i \right\}$
8:     $\mathcal{C}_t = \{a_i \mid \overline{\text{Cpld}}(a_i) = \max_j \overline{\text{Cpld}}(a_j)\}$
9:     Set $\mathcal{B}_t \leftarrow \mathcal{B}_{t-1}$ and $\mathcal{B}_t^i \leftarrow \mathcal{B}_{t-1}^i$ and update as follows:
   **A. Reset disproven hypotheses:** If for any $i$ and $a_j \in \mathcal{B}_t^i$ we have $l_{ij} > 0.5$, reset $\mathcal{B}_t, \overline{L}_C$ and
       $\mathcal{B}_t^k$ for all $k$ (i.e., set them to their original values as in Lines 2–4 above).
   **B. Remove non-Copeland winners:** For each $a_i \in \mathcal{B}_t$, if $\overline{\text{Cpld}}(a_i) < \underline{\text{Cpld}}(a_j)$ holds for any
       $j$, set $\mathcal{B}_t \leftarrow \mathcal{B}_t \setminus \{a_i\}$, and if $|\mathcal{B}_t^i| \neq \overline{L}_C + 1$, then set $\mathcal{B}_t^i \leftarrow \{a_k | u_{ik} < 0.5\}$. However, if
       $\mathcal{B}_t = \varnothing$, reset $\mathcal{B}_t, \overline{L}_C$ and $\mathcal{B}_t^k$ for all $k$.
   **C. Add Copeland winners:** For any $a_i \in \mathcal{C}_t$ with $\overline{\text{Cpld}}(a_i) = \underline{\text{Cpld}}(a_i)$, set $\mathcal{B}_t \leftarrow \mathcal{B}_t \cup \{a_i\}$,
       $\mathcal{B}_t^i \leftarrow \varnothing$ and $\overline{L}_C \leftarrow K - 1 - \overline{\text{Cpld}}(a_i)$. For each $j \neq i$, if we have $|\mathcal{B}_t^j| < \overline{L}_C + 1$, set
       $\mathcal{B}_t^j \leftarrow \varnothing$, and if $|\mathcal{B}_t^j| > \overline{L}_C + 1$, randomly choose $\overline{L}_C + 1$ elements of $\mathcal{B}_t^j$ and remove the rest.
10:     With probability $1/4$, sample $(c, d)$ uniformly from the set $\{(i, j) \mid a_j \in \mathcal{B}_t^i$ and $0.5 \in [l_{ij}, u_{ij}]\}$ (if it is non-empty) and skip to Line 14.
11:     If $\mathcal{B}_t \cap \mathcal{C}_t \neq \varnothing$, then with probability $2/3$, set $\mathcal{C}_t \leftarrow \mathcal{B}_t \cap \mathcal{C}_t$.
12:     Sample $a_c$ from $\mathcal{C}_t$ uniformly at random.
13:     With probability $1/2$, choose the set $\mathcal{B}^i$ to be either $\mathcal{B}_t^i$ or $\{a_1, \ldots, a_K\}$ and then set
       $d \leftarrow \arg\max_{\{j \in \mathcal{B}^i \mid l_{jc} \leq 0.5\}} u_{jc}$. If there is a tie, $d$ is not allowed to be equal to $c$.
14:     Compare arms $a_c$ and $a_d$ and increment $w_{cd}$ or $w_{dc}$ depending on which arm wins.
15: **end for**

---

CCB always chooses $a_d$ from arms of type (3) because comparing $a_c$ and a type (3) arm is most informative about the Copeland score of $a_c$. Among arms of type (3), CCB favors those that have confidently beaten arm $a_c$ in the past (Line 13), i.e., arms that in some round $t' < t$ were of type (2). Such arms are maintained in a shortlist of "formidable" opponents ($\mathcal{B}_t^i$) that are likely to confirm that $a_i$ is not a Copeland winner; these arms are favored when selecting $a_d$ (Lines 10 and 13).

The sets $\mathcal{B}_t^i$ are what speed up the elimination of non-Copeland winners, enabling regret bounds that scale asymptotically with $K$ rather than $K^2$. Specifically, for a non-Copeland winner $a_i$, the set $\mathcal{B}_t^i$ will eventually contain $L_C + 1$ strong opponents for $a_i$ (Line 4.1C), where $L_C$ is the number of losses of each Copeland winner. Since $L_C$ is typically small (cf. Appendix C.3), asymptotically this leads to a bound of only $\mathcal{O}(\log T)$ on the number of time-steps when $a_i$ is chosen as an optimistic Copeland winner, instead of a bound of $\mathcal{O}(K \log T)$, which a more naive algorithm would produce.

## 4.2 Scalable Copeland Bandits (SCB)

SCB is designed to handle dueling bandit problems with large numbers of arms. It is based on an arm-identification algorithm, described in Algorithm 2, designed for a PAC setting, i.e., it finds an $\epsilon$-Copeland winner with probability $1 - \delta$, although we are primarily interested in the case with $\epsilon = 0$. Algorithm 2 relies on a reduction to a $K$-armed bandit problem where we have direct access

---

**Algorithm 2** Approximate Copeland Bandit Solver

---

**Input:** A Copeland dueling bandit problem with preference matrix $\mathbf{P} = [p_{ij}]$, failure probability
   $\delta > 0$, and approximation parameter $\epsilon > 0$. Also, define $[K] := \{1, \ldots, K\}$.
1: Define a random variable $\text{reward}(i)$ for $i \in [K]$ as the following procedure: pick a uniformly
   random $j \neq i$ from $[K]$; query the pair $(a_i, a_j)$ sufficiently many times in order to determine
   w.p. at least $1 - \delta/K^2$ whether $p_{ij} > 1/2$; return 1 if $p_{ij} > 0.5$ and 0 otherwise.
2: Invoke Algorithm 4, where in each of its calls to $\text{reward}(i)$, the feedback is determined by the
   above stochastic process.
**Return:** The same output returned by Algorithm 4.

---

to a noisy version of the Copeland score; the process of estimating the score of arm $a_i$ consists of comparing $a_i$ to a random arm $a_j$ until it becomes clear which arm beats the other. The sample complexity bound, which yields the regret bound, is achieved by combining a bound for $K$-armed bandits and a bound on the number of arms that can have a high Copeland score.

Algorithm 2 calls a $K$-armed bandit algorithm as a subroutine. To this end, we use the KL-based arm-elimination algorithm, a slight modification of Algorithm 2 in [22]: it implements an elimination tournament with confidence regions based on the KL-divergence between probability distributions. The interested reader can find the pseudo-code in Algorithm 4 contained in Appendix J.

Combining this with the *squaring trick*, a modification of the *doubling trick* that reduces the number of partitions from $\log T$ to $\log \log T$, the SCB algorithm, described in Algorithm 3, repeatedly calls Algorithm 2 but force-terminates if an increasing threshold is reached. If it terminates early, then the identified arm is played against itself until the threshold is reached.

---

**Algorithm 3** Scalable Copeland Bandits

---
**Input:** A Copeland dueling bandit problem with preference matrix $\mathbf{P} = [p_{ij}]$
1: **for all** $r = 1, 2, \ldots$ **do**
2:    Set $T = 2^{2^r}$ and run Algorithm 2 with failure probability $\log(T)/T$ in order to find an exact Copeland winner ($\epsilon = 0$); force-terminate if it requires more than $T$ queries.
3:    Let $T_0$ be the number of queries used by invoking Algorithm 2, and let $a_i$ be the arm produced by it; query the pair $(a_i, a_i)$ $T - T_0$ times.
4: **end for**

---

## 5 Theoretical Results

In this section, we present regret bounds for both CCB and SCB. Assuming that the number of Copeland winners and the number of losses of each Copeland winner are bounded,[2] CCB's regret bound takes the form $\mathcal{O}(K^2 + K \log T)$, while SCB's is of the form $\mathcal{O}(K \log K \log T)$. Note that these bounds are not directly comparable. When there are relatively few arms, CCB is expected to perform better. By contrast, when there are many arms SCB is expected to be superior. Appendix A in the supplementary material provides empirical evidence to support these expectations.

Throughout this section we impose the following condition on the preference matrix:

    **A** There are no ties, i.e., for all pairs $(a_i, a_j)$ with $i \neq j$, we have $p_{ij} \neq 0.5$.

This assumption is not very restrictive in practice. For example, in the ranker evaluation setting from information retrieval, each arm corresponds to a ranker, a complex and highly engineered system, so it is unlikely that two rankers are indistinguishable. Furthermore, some of the results we present in this section actually hold under even weaker assumptions. However, for the sake of clarity, we defer a discussion of these nuanced differences to Appendix F in the supplementary material.

### 5.1 Copeland Confidence Bound (CCB)

In this section, we provide a rough outline of our argument for the bound on the regret accumulated by Algorithm 1. For a more detailed argument, the interested reader is referred to Appendix E.

Consider a $K$-armed Copeland bandit problem with arms $a_1, \ldots, a_K$ and preference matrix $\mathbf{P} = [p_{ij}]$, such that arms $a_1, \ldots, a_C$ are the Copeland winners, with $C$ being the number of Copeland winners. Moreover, we define $L_C$ to be the number of arms to which a Copeland winner loses in expectation.

Using this notation, our expected regret bound for CCB takes the form: $\mathcal{O} \left( \frac{K^2 + (C + L_C) K \ln T}{\Delta^2} \right)$   (1)
Here, $\Delta$ is a notion of gap defined in Appendix E, which is an improvement upon the smallest gap between any pair of arms.

This result is proven in two steps. First, we bound the number of comparisons involving non-Copeland winners, yielding a result of the form $\mathcal{O}(K^2 \ln T)$. Second, Theorem 3 closes the gap

between this bound and the one in (1) by showing that, beyond a certain time horizon, CCB selects non-Copeland winning arms as the optimistic Copeland winner very infrequently.

**Theorem 3.** *Given a Copeland bandit problem satisfying Assumption* **A** *and any* $\delta > 0$ *and* $\alpha > 0.5$, *there exist constants* $A_\delta^{(1)}$ *and* $A_\delta^{(2)}$ *such that, with probability* $1 - \delta$, *the regret accumulated by CCB is bounded by the following:*

$$A_\delta^{(1)} + A_\delta^{(2)} \sqrt{\ln T} + \frac{2K(C + L_C + 1)}{\Delta^2} \ln T.$$

Using the high probability regret bound given in Theorem 3, we can deduce the expected regret result claimed in (1) for $\alpha > 1$, as a corollary by integrating $\delta$ over the interval $[0, 1]$.

## 5.2 Scalable Copeland Bandits

We now turn to our regret result for SCB, which lowers the $K^2$ dependence in the additive constant of CCB's regret result to $K \log K$. We begin by defining the relevant quantities:

**Definition 4.** Given a $K$-armed Copeland bandit problem and an arm $a_i$, we define the following:

1. Recall that $\text{cpld}(a_i) := \text{Cpld}(a_i)/(K - 1)$ is called the normalized Copeland score.
2. $a_i$ is an $\epsilon$-Copeland-winner if $1 - \text{cpld}(a_i) \leq (1 - \text{cpld}(a_1))(1 + \epsilon)$.
3. $\Delta_i := \max\{\text{cpld}(a_1) - \text{cpld}(a_i), 1/(K - 1)\}$ and $H_i := \sum_{j \neq i} \frac{1}{\Delta_{ij}^2}$, with $H_\infty := \max_i H_i$.
4. $\Delta_i^\epsilon = \max\{\Delta_i, \epsilon(1 - \text{cpld}(a_1))\}$.

We now state our main scalability result:

**Theorem 5.** *Given a Copeland bandit problem satisfying Assumption* **A***, the expected regret of SCB (Algorithm 3) is bounded by* $\mathcal{O}\left(\frac{1}{K}\sum_{i=1}^K \frac{H_i(1 - \text{cpld}(a_i))}{\Delta_i^2}\right) \log(T)$, *which in turn can be bounded by* $\mathcal{O}\left(\frac{K(L_C + \log K)\log T}{\Delta_{\min}^2}\right)$, *where* $L_C$ *and* $\Delta_{\min}$ *are as in Definition 10.*

Recall that SCB is based on Algorithm 2, an arm-identification algorithm that identifies a Copeland winner with high probability. As a result, Theorem 5 is an immediate corollary of Lemma 6, obtained by using the well known squaring trick. As mentioned in Section 4.2, the squaring trick is a minor variation on the doubling trick that reduces the number of partitions from $\log T$ to $\log \log T$.

Lemma 6 is a result for finding an $\epsilon$-approximate Copeland winner (see Definition 4.2). Note that, for the regret setting, we are only interested in the special case with $\epsilon = 0$, i.e., the problem of identifying the best arm.

**Lemma 6.** *With probability* $1 - \delta$, *Algorithm 2 finds an* $\epsilon$-*approximate Copeland winner by time*

$$\mathcal{O}\left(\frac{1}{K}\sum_{i=1}^K \frac{H_i(1 - \text{cpld}(a_i))}{(\Delta_i^\epsilon)^2}\right) \log(1/\delta) \leq \mathcal{O}\left(H_\infty\left(\log(K) + \min\{\epsilon^{-2}, L_C\}\right)\right) \log(1/\delta).$$

*assuming[3]* $\delta = (KH_\infty)^{\Omega(1)}$. *In particular when there is a Condorcet winner* ($\text{cpld}(a_1) = 1, L_C = 0$) *or more generally* $\text{cpld}(a_1) = 1 - \mathcal{O}(1/K), L_C = \mathcal{O}(1)$, *an exact solution is found with probability at least* $1 - \delta$ *by using an expected number of queries of at most* $\mathcal{O}(H_\infty(L_C + \log K)) \log(1/\delta)$.

In the remainder of this section, we sketch the main ideas underlying the proof of Lemma 6, detailed in Appendix I in the supplementary material. We first treat the simpler deterministic setting in which a single query suffices to determine which of a pair of arms beats the other. While a solution can easily be obtained using $K(K - 1)/2$ many queries, we aim for one with query complexity linear in $K$. The main ingredients of the proof are as follows:

1. $\text{cpld}(a_i)$ is the mean of a Bernoulli random variable defined as such: sample uniformly at random an index $j$ from the set $\{1, \ldots, K\} \setminus \{i\}$ and return 1 if $a_i$ beats $a_j$ and 0 otherwise.
2. Applying a KL-divergence based arm-elimination algorithm (Algorithm 4) to the $K$-armed bandit arising from the above observation, we obtain a bound by dividing the arms into two groups: those with Copeland scores close to that of the Copeland winners, and the rest. For the former, we use the result from Lemma 7 to bound the number of such arms; for the latter, the resulting regret is dealt with using Lemma 8, which exploits the possible distribution of Copeland scores.

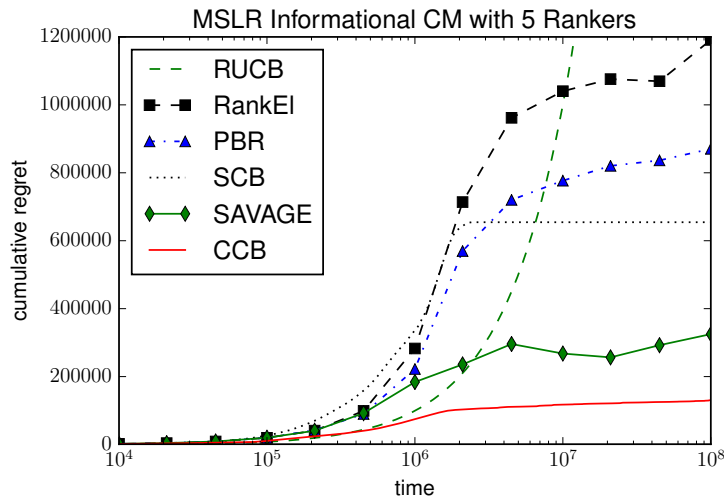

Figure 1: Small-scale regret results for a 5-armed Copeland dueling bandit problem arising from ranker evaluation.

Let us state the two key lemmas here:

**Lemma 7.** *Let $D \subset \{a_1, \ldots, a_K\}$ be the set of arms for which $\mathrm{cpld}(a_i) \geq 1 - d/(K-1)$, that is arms that are beaten by at most $d$ arms. Then $|D| \leq 2d + 1$.*

*Proof.* Consider a fully connected directed graph, whose node set is $D$ and the arc $(a_i, a_j)$ is in the graph if arm $a_i$ beats arm $a_j$. By the definition of cpld, the in-degree of any node $i$ is upper bounded by $d$. Therefore, the total number of arcs in the graph is at most $|D|d$. Now, the full connectivity of the graph implies that the total number of arcs in the graph is exactly $|D|(|D| - 1)/2$. Thus, $|D|(|D| - 1)/2 \leq |D|d$ and the claim follows. ☐

**Lemma 8.** *The sum $\sum_{\{i|\mathrm{cpld}(a_i)<1\}} \frac{1}{1-\mathrm{cpld}(a_i)}$ is in $\mathcal{O}(K \log K)$.*

*Proof.* Follows from Lemma 7 via a careful partitioning of arms. Details are in Appendix I. ☐

Given the structure of Algorithm 2, the stochastic case is similar to the deterministic case for the following reason: while the latter requires a single comparison between arms $a_i$ and $a_j$ to determine which arm beats the other, in the stochastic case, we need roughly $\frac{\log(K \log(\Delta_{ij}^{-1})/\delta)}{\Delta_{ij}^2}$ comparisons between the two arms to correctly answer the same question with probability at least $1 - \delta/K^2$.

## 6 Experiments

To evaluate our methods CCB and SCB, we apply them to a Copeland dueling bandit problem arising from *ranker evaluation* in the field of *information retrieval* (IR) [23].

We follow the experimental approach in [3, 13] and use a preference matrix to simulate comparisons between each pair of arms $(a_i, a_j)$ by drawing samples from Bernoulli random variables with mean $p_{ij}$. We compare our proposed algorithms against the state of the art $K$-armed dueling bandit algorithms, RUCB [13], Copeland SAVAGE, PBR and RankEl. We include RUCB in order to verify our claim that $K$-armed dueling bandit algorithms that assume the existence of a Condorcet winner have linear regret if applied to a Copeland dueling bandit problem without a Condorcet winner.

More specifically, we consider a 5-armed dueling bandit problem obtained from comparing five rankers, none of whom beat the other four, i.e. there is no Condorcet winner. Due to lack of space, the details of the experimental setup have been included in Appendix B[4]. Figure 1 shows the regret accumulated by CCB, SCB, the Copeland variants of SAVAGE, PBR, RankEl and RUCB on this problem. The horizontal time axis uses a log scale, while the vertical axis, which measures cumulative regret, uses a linear scale. CCB outperforms all other algorithms in this 5-armed experiment.

Note that three of the baseline algorithms under consideration here (i.e., SAVAGE, PBR and RankEl) require the horizon of the experiment as an input, either directly or through a failure probability $\delta$,

which we set to $1/T$ (with $T$ being the horizon), in order to obtain a finite-horizon regret algorithm, as prescribed in [3, 10]. Therefore, we ran independent experiments with varying horizons and recorded the accumulated regret: the markers on the curves corresponding to these algorithms represent these numbers. Consequently, the regret curves are not monotonically increasing. For instance, SAVAGE's cumulative regret at time $2 \times 10^7$ is lower than at time $10^7$ because the runs that produced the former number were not continuations of those that resulted in the latter, but rather completely independent. Furthermore, RUCB's cumulative regret grows linearly, which is why the plot does not contain the entire curve.

Appendix A contains further experimental results, including those of our scalability experiment.

# 7 Conclusion

In many applications that involve learning from human behavior, feedback is more reliable when provided in the form of pairwise preferences. In the dueling bandit problem, the goal is to use such pairwise feedback to find the most desirable choice from a set of options. Most existing work in this area assumes the existence of a Condorcet winner, i.e., an arm that beats all other arms with probability greater than $0.5$. Even though these results have the advantage that the bounds they provide scale linearly in the number of arms, their main drawback is that in practice the Condorcet assumption is too restrictive. By contrast, other results that do not impose the Condorcet assumption achieve bounds that scale quadratically in the number of arms.

In this paper, we set out to solve a natural generalization of the problem, where instead of assuming the existence of a Condorcet winner, we seek to find a Copeland winner, which is guaranteed to exist. We proposed two algorithms to address this problem: one for small numbers of arms, called CCB, and a more scalable one, called SCB, that works better for problems with large numbers of arms. We provided theoretical results bounding the regret accumulated by each algorithm: these results improve substantially over existing results in the literature, by filling the gap that exists in the current results, namely the discrepancy between results that make the Condorcet assumption and are of the form $\mathcal{O}(K \log T)$ and the more general results that are of the form $\mathcal{O}(K^2 \log T)$.

Moreover, we have included in the supplementary material empirical results on both a dueling bandit problem arising from a real-life application domain and a large-scale synthetic problem used to test the scalability of SCB. The results of these experiments show that CCB beats all existing Copeland dueling bandit algorithms, while SCB outperforms CCB on the large-scale problem.

One open question raised by our work is how to devise an algorithm that has the benefits of both CCB and SCB, i.e., the scalability of the latter together with the former's better dependence on the gaps. At this point, it is not clear to us how this could be achieved. Another interesting direction for future work is an extension of both CCB and SCB to problems with a continuous set of arms. Given the prevalence of cyclical preference relationships in practice, we hypothesize that the non-existence of a Condorcet winner is an even greater issue when dealing with an infinite number of arms. Given that both our algorithms utilize confidence bounds to make their choices, we anticipate that continuous-armed UCB-style algorithms like those proposed in [24, 25, 26, 27, 28, 29, 30] can be combined with our ideas to produce a solution to the continuous-armed Copeland bandit problem that does not rely on the convexity assumptions made by algorithms such as the one proposed in [31]. Finally, it is also interesting to expand our results to handle scores other than the Copeland score, such as an $\epsilon$-insensitive variant of the Copeland score (as in [12]), or completely different notions of winners, such as the Borda, Random Walk or von Neumann winners (see, e.g., [32, 19]).

**Acknowledgments**

We would like to thank Nir Ailon and Ulle Endriss for helpful discussions. This research was supported by Amsterdam Data Science, the Dutch national program COMMIT, Elsevier, the European Community's Seventh Framework Programme (FP7/2007-2013) under grant agreement nr 312827 (VOX-Pol), the ESF Research Network Program ELIAS, the Royal Dutch Academy of Sciences (KNAW) under the Elite Network Shifts project, the Microsoft Research Ph.D. program, the Netherlands eScience Center under project number 027.012.105, the Netherlands Institute for Sound and Vision, the Netherlands Organisation for Scientific Research (NWO) under project nrs 727.011.005, 612.001.116, HOR-11-10, 640.006.013, 612.066.930, CI-14-25, SH-322-15, the Yahoo! Faculty Research and Engagement Program, and Yandex. All content represents the opinion of the authors, which is not necessarily shared or endorsed by their respective employers and/or sponsors.

## Footnotes

[1]Cf. [10, Equation 9 in §4.1.1] and [11, Theorem 1].

[2]See Appendix C.3 in the supplementary material for experimental evidence that this is the case in practice.

[3]The exact expression requires replacing $\log(1/\delta)$ with $\log(KH_\infty/\delta)$.

[4]Sample code and the preference matrices used in the experiments can be found at http://bit.ly/nips15data.

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
