[Supplementary Material]

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

[5]See Tables 2 and 3 in the supplementary material for a summary of the definitions used in this paper.

[6]See Figures 7 and 8 in the supplementary material for a pictorial explanation.

[7]The value of $\delta$ we require is $1/T$. If the assumption does not follow in that case, the regret must be linear and all of the statements hold trivially.

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

# Appendix

## A  Experimental Results

In this section, we continue using the experimental setup laid out in Section 6 to carry out a more detailed investigation of our proposed algorithms. In particular, we conduct both a scalability experiment to understand the behaviours of CCB and SCB as the number of arms grows as well as an experiment on a dueling bandit problem that satisfies the Condorcet assumption.

Figure 2: Large-scale regret results for a synthetic 500-armed Copeland dueling bandit problem.

Our scalability experiment uses a $500$-armed synthetic example created to test the scalability of SCB. In particular, we fix a preference matrix in which the three Copeland winners are in a cycle, each with a Copeland score of 498, and the other arms have Copeland scores ranging from 0 to 496.

Figure 2, which depicts the results of this experiment, shows that when there are many arms, SCB can substantially outperform CCB. We omit SAVAGE, PBR and RankEl from this experiment because they scale poorly in the number of arms [10, 11, 12].

The reason for the sharp transition in the regret curves of CCB and SCB in the synthetic experiment is as follows. Because there are many arms, as long as one of the two arms being compared is not a Copeland winner, the comparison can result in substantial regret; since both algorithms choose the second arm in each round based on some criterion other than the Copeland score, even if the first chosen arm in a given time-step is a Copeland winner, the incurred regret may be as high as $0.5$. The sudden transition in Figure 2 occurs when the algorithm becomes confident enough of its choice for the first arm to begin comparing it against itself, at which point it stops accumulating regret.

As advertised previously, our next experiment is on an example with a Condorcet winner in order to show how CCB compares against RUCB when the condition required by RUCB is satisfied. The regret plots for the remaining algorithms were excluded here since they both perform substantially worse than either RUCB or CCB, as expected. This example was extracted in the same fashion as the example used in the ranker evaluation experiment detailed in Appendix B, with the sole difference that this time we ensured that one of the rankers is a Condorcet winner. The results, depicted in Figure 3, show that CCB enjoys a slight advantage over RUCB in this case. We attribute this to the careful process of identifying and utilizing the weaknesses of non-Copeland winners, as carried out by lines 12 and 18 of Algorithm 1.

Figure 3: Regret results for a Condorcet example.

## B   Ranker Evaluation Details

A ranker is a function that takes as input a user's search query and ranks the documents in a collection according to their relevance to that query. Ranker evaluation aims to determine which among a set of rankers performs best. One effective way to achieve this is to use *interleaved comparisons* [33], which interleave the ranked lists of documents proposed by two rankers and present the resulting list to the user, whose subsequent click feedback is used to infer a noisy preference for one of the rankers. Given a set of $K$ rankers, the problem of finding the best ranker can then be modeled as a $K$-armed dueling bandit problem, with each arm corresponding to a ranker.

We use interleaved comparisons to estimate the preference matrix for the full set of rankers included with the MSLR dataset [34], from which we select 5 rankers such that a Condorcet winner does not exist. The MSLR dataset [34] consists of relevance judgments provided by expert annotators assessing the relevance of a given document to a given query. Using this data set, we create a set of 136 rankers, each corresponding to a ranking feature provided in the data set, e.g., PageRank. The ranker evaluation task in this context corresponds to determining which single feature constitutes the best ranker [4].

To compare a pair of rankers, we use *probabilistic interleave* (PI) [35], a recently developed method for interleaved comparisons. To model the user's click behavior on the resulting interleaved lists, we employ a probabilistic user model [35, 36] that uses as input the manual labels (classifying documents as relevant or not for given queries) provided with the MSLR dataset. Queries are sampled randomly and clicks are generated probabilistically by conditioning on these assessments in a way that resembles the behavior of an actual user [37]. Specifically, we employ an informational click model in our ranker evaluation experiments [38].

The informational click model simulates the behavior of users whose goal is to acquire knowledge about multiple facets of a topic, rather than seeking a specific page that contains all the information that they need. As such, in the informational click model, the user tends to continue examining documents even after encountering a highly relevant document. The informational click model is one of the three click models utilized in the ranker evaluation literature, along with the perfect and navigational click models [38]. It turns out that the full preference matrix of the feature vectors of the MSLR dataset has a Condorcet winner when the perfect or the navigational click-models are used. As we will see in Appendix C.1, using the informational click model that is no longer true.

Following [3, 13], we first use the above approach to estimate the comparison probabilities $p_{ij}$ for each pair of rankers and then use these probabilities to simulate comparisons between rankers. More specifically, we estimate the full preference matrix, called the *informational preference matrix*, by performing $400,000$ interleaved comparisons on each pair of the 136 feature rankers.

Figure 4: The probability that the Condorcet assumption holds for subsets of the feature rankers. The probability is shown as a function of the size of the subset.

# C Assumptions and Key Quantities

In this section, we provide quantitative analysis of the various assumptions, definitions and quantities that were discussed in the main body of the paper.

## C.1 The Condorcet Assumption

To test how stringent the Condorcet assumption is, we use the informational preference matrix described in Section B to estimate for each $K = 1, \ldots, 136$ the probability $P_K$ that a given $K$-armed dueling bandit problem, obtained from considering $K$ of our 136 feature rankers, would have a Condorcet winner by randomly selecting $10,000$ $K$-armed dueling bandit problems and counting the ones with Condorcet winners. As can be seen from Figure 4, as $K$ grows the probability that the Condorcet assumption holds decreases rapidly. We hypothesize that this is because the informational click model explores more of the list of ranked documents than the navigational click model, which was used in [13], and so it is more likely to encounter non-transitivity phenomena of the sort described in [39].

## C.2 Other Notions of Winners

As mentioned in Section 3, numerous other definitions of what constitutes the best arm have been proposed, some of which specialize to the Condorcet winner, when it exists. This latter property is desirable both in preference learning and social choice theory: the Condorcet winner is the choice that is preferred over all other choices, so if it exists, there is good reason to insist on selecting it. The Copeland winner, as discussed in this paper, and the von Neumann winner [19, 40] satisfy this property, while the Borda (a.k.a. Sum of Expectations) and the Random Walk (a.k.a. PageRank) winners [41] do not. The von Neumann winner is in fact defined as a distribution over arms such that playing it will maximize the probability to beat any fixed arm. The Borda winner is defined as the arm maximizing the score $\sum_{j \neq i} p_{ij}$ and can be interpreted as the arm that beats other arms by the most, rather than beating the most arms. The Random Walk winner is defined as the arm we are most likely to visit in some Markov Chain determined by the preference matrix. In this section, we provide some numerical evidence for the similarity of these notions in practice, based on the sampled preference matrices obtained from the ranker evaluation from IR, which was described in the Section B/C.1. Table 1 lists the percentage of preference matrices for which pairs of winners overlap. In the case of the von Neumann winner, which is defined as a probability distribution over the set of arms [19], we used the support of the distribution (i.e., the set of arms with non-zero probability) to define overlap with the other definitions.

Table 1: Percentage of matrices for which the different notions of winners overlap in the experimental setup described in Appendices B and C.1.

| Overlap | Copeland | von Neumann | Borda | Random Walk |
|---|---|---|---|---|
| Copeland | 100% | 99.94% | 51.49% | 56.15% |
| von Neumann | 99.94% | 100% | 77.66% | 82.11% |
| Borda | 51.49% | 77.66% | 100% | 94.81% |
| RandomWalk | 56.15% | 82.11% | %94.81 | 100% |

As these numbers demonstrate, the Copeland and the von Neumann winners are very likely to overlap, as are the Borda and Random Walk winners, while the first two definitions are more likely to be incompatible with the latter two. Furthermore, in the case of 94.2% of the preference matrices, all Copeland winners were contained in the support of the von Neumann winner, suggesting that in practice the Copeland winner is a more restrictive notion of what constitutes a winner.

## C.3 The Quantities $C$ and $L_C$

We also examine additional quantities relevant to our regret bounds: the number of Copeland winners, $C$; the number of losses of each Copeland winner, $L_C$; and the range of values in which these quantities fall. Using the above randomly chosen preference sub-matrices, we counted the number of times each possible value for $C$ and $L_C$ was observed. The results are depicted in Figure 5: the area of the circle with coordinates $(x, y)$ is proportional to the percentage of examples with $K = x$

Figure 5: Observed values of the parameters $C$ and $L_C$: the area of the circle with coordinates $(x, y)$ is proportional to the percentage of examples with $K = x$ which satisfied $C = y$ (in the top plot) or $L_C = y$ in the bottom plot.

which satisfied $C = y$ (in the top plot) or $L_C = y$ (in the bottom plot). As these plots show, the parameters $C$ and $L_C$ are generally much lower than $K$.

### C.4 The Gap $\Delta$

The regret bound for CCB, given in (1), depends on the gap $\Delta$ defined in Definition 10.6, rather than the smallest gap $\Delta_{\min}$ as specified in Definition 10.2. The latter would result in a looser regret bound and Figure 6 quantifies this deterioration in the ranker evaluation example under consideration here. In particular, the plot depicts the average of the ratio between the two bounds (the one using $\Delta$ and the one using $\Delta_{\min}$) across the $10,000$ sampled preference matrices used in the analysis of the Condorcet winner for each $K$ in the set $\{2, \ldots, 135\}$. The average ratio decreases as the number of arms approaches 136 because, as $K$ increases, the sampled preference matrices increasingly resemble the full preference matrix and so their gaps $\Delta$ and $\Delta_{\min}$ approach those of the full 136-armed preference matrix as well. As it turns out, the ratio $\Delta^2/\Delta_{\min}^2$ for the full matrix is equal to $1,419$. Hence, the curve in Figure 6 approaches that number as the number of arms approaches 136.

## D Background Material

**Maximal Azuma-Hoeffding Bound** [42, §A.1.3]: Given random variables $X_1, \ldots, X_N$ with common range $[0, 1]$ satisfying $\mathbf{E}[X_n | X_1, \ldots, X_{n-1}] = \mu$, define the partial sums $S_n = X_1 + \cdots + X_n$. Then, for all $a > 0$, we have

$$P\Big( \max_{n \leq N} S_n > n\mu + a \Big) \leq e^{-2a^2/N}$$

$$P\Big( \min_{n \leq N} S_n < n\mu - a \Big) \leq e^{-2a^2/N}$$

Here, we will quote a useful Lemma that we will refer to repeatedly in our proofs:

**Lemma 9** (Lemma 1 in [13]). *Let $\mathbf{P} := [p_{ij}]$ be the preference matrix of a $K$-armed dueling bandit problem with arms $\{a_1, \ldots, a_K\}$. Then, for any dueling bandit algorithm and any $\alpha > \frac{1}{2}$ and $\delta > 0$, we have*

$$P\Big( \forall t > C(\delta), i, j, \ p_{ij} \in [l_{ij}(t), u_{ij}(t)] \Big) > 1 - \delta.$$

Figure 6: The average advantage gained by having the bound in (1) depend on $\Delta$ rather than $\Delta_{\min}$: for each number of arms $K$, the expectation is taken across the $10,000$ $K$-armed preference matrices obtained using the sampling procedure described above.

## E An Outline of the Proof of Theorem 3

To analyze Algorithm 1, consider a $K$-armed Copeland bandit problem with arms $a_1, \dots, a_K$ and preference matrix $\mathbf{P} = [p_{ij}]$, such that arms $a_1, \dots, a_C$ are the Copeland winners, with $C$ being the number of Copeland winners. Throughout this section, we assume that the parameter $\alpha$ in Algorithm 1 satisfies $\alpha > 0.5$, unless otherwise stated. We first define the relevant quantities:

**Definition 10.** Given the above setting we define:[5]
1. $\mathcal{L}_i := \{a_j \mid p_{ij} < 0.5\}$, i.e., the arms to which $a_i$ loses, and $L_C := |\mathcal{L}_1|$.
2. $\Delta_{ij} := |p_{ij} - 0.5|$ and $\Delta_{\min} := \min_{i \neq j} \Delta_{ij}$
3. Given $i > C$, define $i^*$ as the index of the $(L_C + 1)^{th}$ largest element in the set $\{\Delta_{ij} \mid p_{ij} < 0.5\}$.
4. Define $\Delta_i^*$ to be $\Delta_{ii^*}$ if $i > C$ and $0$ otherwise. Moreover, let us set $\Delta_{\min}^* := \min_{i > C} \Delta_i^*$.
5. Define $\Delta_{ij}^*$ to be $\Delta_i^* + \Delta_{ij}$ if $p_{ij} \geq 0.5$ and $\max\{\Delta_i^*, \Delta_{ij}\}$ otherwise.[6]
6. $\Delta := \min\{\min_{i \leq C < j} \Delta_{ij}, \Delta_{\min}^*\}$, where $\Delta_{\min}^*$ is defined as in item 4 above.
7. $C(\delta) := \left((4\alpha - 1)K^2/(2\alpha - 1)\delta\right)^{\frac{1}{2\alpha-1}}$ where $\alpha$ is as in Algorithm 1.
8. $N_{ij}^\delta(t)$ is the number of time-steps between times $C(\delta)$ and $t$ when $a_i$ was chosen as the optimistic Copeland winner and $a_j$ as the challenger. Also, $\widehat{N}_{ij}^\delta(t)$ is defined to be $(4\alpha \ln t)/\left(\Delta_{ij}^*\right)^2$ if $i \neq j$, $0$ if $i = j > C$ and $t$ if $i = j \leq C$. We also define $\widehat{N}^\delta(t) := \sum_{i \neq j} \widehat{N}_{ij}^\delta(t) + 1$.

Using this notation, our expected regret bound for CCB takes the form: $\mathcal{O}\left(\frac{K^2 + (C + L_C)K \ln T}{\Delta^2}\right)$ (2)

This result is proven in two steps. First, Proposition 11 bounds the number of comparisons involving non-Copeland winners, yielding a result of the form $\mathcal{O}(K^2 \ln T)$. Second, Theorem 18 closes the gap between this bound and that of (2) by showing that, beyond a certain time horizon, CCB selects non-Copeland winning arms as the optimistic Copeland winner very infrequently.

Note that we have $\Delta_{ij}^* \geq \Delta_{ij}$ for all pairs $i \neq j$. Thus, for simplicity, the analysis in this section can be read as if the bounds were given in terms of $\Delta_{ij}$. We use $\Delta_{ij}^*$ instead because it gives

tighter upper bounds. In particular, simply using the gaps $\Delta_{ij}$ would replace the denominator of the expression in (2) with $\Delta_{\min}^2$, which leads to a substantially worse regret bound in practice. For instance, in the ranker evaluation application used in the experiments in the supplementary material, this change would on average increase the regret bound by a factor that is of the order of tens of thousands. See Appendix C.4 for a more quantitative discussion of this point.

We can now state our first bound, proved in Appendix F under weaker assumptions.

**Proposition 11.** *Given any $\delta > 0$ and $\alpha > 0.5$, if we apply CCB (Algorithm 1) to a dueling bandit problem satisfying Assumption **A**, the following holds with probability $1 - \delta$: for any $T > C(\delta)$ and any pair of arms $a_i$ and $a_j$, we have $N_{ij}^\delta(T) \leq \widehat{N}_{ij}^\delta(T)$.*

One can sum the inequalities in the last proposition over pairs $(i, j)$ to get a regret bound of the form $\mathcal{O}(K^2 \log T)$ for Algorithm 1. However, as Theorem 18 will show, we can use the properties of the sets $\mathcal{B}_t^i$ to obtain a tighter regret bound of the form $\mathcal{O}(K \log T)$. Before stating that theorem, we need a few definitions and lemmas. We begin by defining the key quantity:

**Definition 12.** Given a preference matrix $\mathbf{P}$ and $\delta > 0$, then $T_\delta$ is the smallest integer satisfying
$$T_\delta \geq C(\tfrac{\delta}{2}) + 8K^2(L_C+1)^2 \ln \tfrac{6K^2}{\delta} + K^2 \ln \tfrac{6K}{\delta} + \tfrac{32\alpha K(L_C+1)}{\Delta_{\min}^2} \ln T_\delta + \widehat{N}^{\frac{\delta}{2}}(T_\delta) + 4K \max_{i>C} \widehat{N}_i^{\frac{\delta}{2}}(T_\delta).$$

**Remark 13.** $T_\delta$ is $\text{poly}(K, \delta^{-1})$ and our regret bound below scales as $\log T_\delta$.

The following two lemmas are key to the proof of Theorem 18. Lemma 14 (proved in Appendix G) states that, with high probability by time $T_\delta$, each set $\mathcal{B}_t^i$ contains $L_C + 1$ arms $a_j$, each of which beats $a_i$ (i.e., $p_{ij} < 0.5$). This fact then allows us to prove Lemma 15 (Appendix H), which states that, after time-step $T_\delta$, the rate of suboptimal comparisons is $\mathcal{O}(K \ln T)$ rather than $\mathcal{O}(K^2 \ln T)$.

**Lemma 14.** *Given $\delta > 0$, with probability $1 - \delta$, each set $\mathcal{B}_{T_\delta}^i$ with $i > C$ contains exactly $L_C + 1$ elements with each element $a_j$ satisfying $p_{ij} < 0.5$. Moreover, for all $t \in [T_\delta, T]$, we have $\mathcal{B}_t^i = \mathcal{B}_{T_\delta}^i$.*

**Lemma 15.** *Given a Copeland bandit problem satisfying Assumption **A** and any $\delta > 0$, with probability $1 - \delta$ the following holds: the number of time-steps between $T_{\delta/2}$ and $T$ when each non-Copeland winner $a_i$ can be chosen as optimistic Copeland winners (i.e., times when arm $a_c$ in Algorithm 1 satisfies $c > C$) is bounded by $\widehat{N}^i := 2\widehat{N}_\mathcal{B}^i + 2\sqrt{\widehat{N}_\mathcal{B}^i \ln \tfrac{2K}{\delta}}$, where $\widehat{N}_\mathcal{B}^i := \sum_{j \in \mathcal{B}_{T_{\delta/2}}^i} \widehat{N}_{ij}^{\delta/4}(T)$.*

**Remark 16.** Due to Lemma 14, with high probability we have $\widehat{N}_\mathcal{B}^i \leq \tfrac{(L_C+1)\ln T}{(\Delta_{\min}^*)^2}$ for each $i > C$ and so the total number of times between $T_\delta$ and $T$ when a non-Copeland winner is chosen as an optimistic Copeland winner is in $\mathcal{O}(KL_C \ln T)$ for a fixed minimal gap $\Delta_{\min}^*$. The only other way a suboptimal comparison can occur is if a Copeland winner is compared against a non-Copeland winner, and according to Proposition 11, the number of such occurrences is bounded by $\mathcal{O}(KC \ln T)$. Hence, the number of suboptimal comparisons is in $\mathcal{O}(K \ln T)$ assuming that $C$ and $L_C$ are bounded. In Appendix C.3 in the supplementary material, we provide experimental evidence for this.

We now define the quantities needed to state the main theorem.

**Definition 17.** We define the following three quantities: $A_\delta^{(1)} := C(\delta/4) + \widehat{N}^\delta(T_{\delta/2})$, $A_\delta^{(2)} := \sum_{i>C} \tfrac{\sqrt{L_C+1}}{\Delta_i^*} \ln \tfrac{2K}{\delta}$ and $A^{(3)} := \sum_{i \leq C < j} \tfrac{1}{(\Delta_{ij})^2} + 2 \sum_{i>C} \tfrac{L_C+1}{(\Delta_i^*)^2}$.

Finally, we repeat the statement of Theorem 3 for the reader's convenience.

**Theorem 18.** *Given a Copeland bandit problem satisfying Assumption **A** and any $\delta > 0$ and $\alpha > 0.5$, with probability $1 - \delta$, the regret accumulated by CCB is bounded by the following:*

$$A_\delta^{(1)} + A_\delta^{(2)}\sqrt{\ln T} + A^{(3)} \ln T \;\leq\; A_\delta^{(1)} + A_\delta^{(2)}\sqrt{\ln T} + \frac{2K(C + L_C + 1)}{\Delta^2} \ln T.$$

For a general assessment of the above quantities, assuming that $L_C$ and $C$ are both $\mathcal{O}(1)$, the above quantities in terms of $K$ become $A_\delta^{(1)} = \mathcal{O}(K^2)$, $A_\delta^{(2)} = \mathcal{O}(K \log(K))$, $A^{(3)} = \mathcal{O}(K)$. Hence, the above bound boils down to the expression in (2). We now turn to the proof of the theorem.

*Proof of Theorem 18.* Let us consider the two disjoint time-intervals $[1, T_{\delta/2}]$ and $(T_{\delta/2}, T]$:

**[1,T$_{\delta/2}$]:** In this case, applying Proposition 11 to $T_\delta$, we get that the number of time-steps when a non-Copeland winner was compared against another arm is bounded by $A_\delta^{(1)}$. As the maximum regret such a comparison can incur is 1, this deals with the first term in the above expression.
**(T$_{\delta/2}$,T]:** In this case, applying Lemma 15, we get the other two terms in the above regret bound.
$\square$

Now that we have the high probability regret bound given in Theorem 18, we can deduce the expected regret result claimed in (2) for $\alpha > 1$, as a corollary by integrating $\delta$ over the interval $[0, 1]$.

## F  Proof of Proposition 11

Before starting with the proof, let us point out the following two properties that can be derived from Assumption **A** in Section 5:

**P1** There are no ties involving a Copeland winner and a non-Copeland winner, i.e., for all pairs of arms $(a_i, a_j)$ with $i \leq C < j$, we have $p_{ij} \neq 0.5$.
**P2** Each non-Copeland winner has more losses than every Copeland winner, i.e., for every pair of arms $(a_i, a_j)$, with $i \leq C < j$, we have $|\mathcal{L}_i| < |\mathcal{L}_j|$.

Even though we have assumed in the statement of Proposition 11 that Assumption **A** holds, it turns out that the proof provided in this section holds as long as the above two properties hold.

**Proposition 11** *Applying CCB to a dueling bandit problem satisfying properties **P1** and **P2**, we have the following bounds on the number of comparisons involving various arms for each $T > C(\delta)$: for each pair of arms $a_i$ and $a_j$, such that either at least one of them is not a Copeland winner or $p_{ij} \neq 0.5$, with probability $1 - \delta$ we have*

$$N_{ij}^\delta(T) \leq \widehat{N}_{ij}^\delta(T) := \begin{cases} \dfrac{4\alpha \ln T}{\left(\Delta_{ij}^*\right)^2} & \text{if } i \neq j \\ 0 & \text{if } i = j > C \end{cases} \tag{3}$$

*Proof of Proposition 11.*  We will prove these bounds by considering a number of cases separately:

1. $i \leq C$ **and** $p_{ij} \neq 0.5$: First of all, since $a_i$ is a Copeland winner, this means that according to the definitions in Tables 2 and 3, $\Delta_{ij}^*$ is simply equal to $\Delta_{ij}$; secondly, assuming by way of contradiction that $N_{ij}^\delta(t) > \frac{4\alpha \ln T}{\Delta_{ij}} > 0$, then we have $\tau_{ij} > C(\delta)$ and so by Lemma 9, we have with probability $1 - \delta$ that the confidence interval $[l_{ij}(\tau_{ij}), u_{ij}(\tau_{ij})]$ contains the preference probability $p_{ij}$. But, in order for arm $a_j$ to have been chosen as the challenger to $a_i$, we must also have $0.5 \in [l_{ij}(\tau_{ij}), u_{ij}(\tau_{ij})]$; to see this, let us consider the two possible cases:

   (a) If we have $p_{ij} > 0.5$, then having

   $$0.5 \notin [l_{ij}(\tau_{ij}), u_{ij}(\tau_{ij})]$$

   implies that we have $l_{ij}(\tau_{ij}) > 0.5$, which in turn implies

   $$u_{ji}(\tau_{ij}) = 1 - l_{ij}(\tau_{ij}) < 0.5 = u_{ii}(\tau_{ij}),$$

   but this is impossible since in that case $a_i$ would've been chosen as the challenger.
   (b) If we have $p_{ij} < 0.5$, then have

   $$0.5 \notin [l_{ij}(\tau_{ij}), u_{ij}(\tau_{ij})]$$

   implies that we have $u_{ij}(\tau_{ij}) < 0.5$, but this is impossible because it means that we had $l_{ji}(\tau_{ij}) > 0.5$, and CCB would've eliminated it from considerations in its second round.

   So, in either case, we cannot have $0.5 \notin [l_{ij}(\tau_{ij}), u_{ij}(\tau_{ij})]$. Therefore, at time $\tau_{ij}$, we must have had $u_{ij}(\tau_{ij}) - l_{ij}(\tau_{ij}) > |p_{ij} - 0.5| =: \Delta_{ij}$. From this, we can conclude the following, using

Figure 7: This figure illustrates the definition of the quantities $\Delta_i^*$ and $\Delta_{ij}^*$ in the case that arm $a_i$ is a Copeland winner, as well as the idea behind Case 1 in the proof of Proposition 11. In this setting we have $\Delta_i^* = 0$ and $\Delta_{ij}^* = \Delta_{ij}$. On the one hand, by Lemma 9, we know that the confidence intervals will contain the $p_{ij}$ (the blue dots in the plots), and on the other as soon as the confidence interval of $p_{ij}$ stops containing $0.5$ for some arm $a_j$, we know that it could not be chosen to be compared against $a_i$. In this way, the gaps $\Delta_{ij}^*$ regulate the number of times that arm each arm can be chosen to be played against $a_i$ during time-steps when $a_i$ is chosen as optimistic Copeland winner.

the definition of $u_{ij}$ and $l_{ij}$:

$$u_{ij}(\tau_{ij}) - l_{ij}(\tau_{ij}) := 2\sqrt{\frac{\alpha \ln \tau_{ij}}{N_{ij}(\tau_{ij})}} \geq \Delta_{ij}$$

$$\therefore \quad 2\sqrt{\frac{\alpha \ln \tau_{ij}}{N_{ij}^\delta(\tau_{ij})}} \geq \Delta_{ij} \quad \because N_{ij}^\delta(\tau_{ij}) \leq N_{ij}(\tau_{ij})$$

$$\therefore \quad 2\sqrt{\frac{\alpha \ln T}{N_{ij}^\delta(\tau_{ij})}} \geq \Delta_{ij} \quad \because \tau_{ij} \leq T$$

$$\therefore \quad N_{ij}^\delta(\tau_{ij}) \leq \frac{4\alpha \ln T}{\Delta_{ij}^2},$$

giving us the desired bound. The reader is referred to Figure 7 for an illustration of this argument.

2. $C < i$: Let us deal with the two cases included in Inequality (3) separately:

   (a) $i = j > C$: In plain terms, this says that with probability $1 - \delta$ no non-Copeland winner will be compared against itself after time $C(\delta)$. The reason for this is the following set of facts:
   - Since $a_i$ is a non-Copeland winner, we have by Property **P1** that it loses to more arms than any Copeland winner.
   - For $a_i$ to have been chosen as an optimistic Copeland winner, it has to have (optimistically) lost to no more than $L_C$ arms, which means that there exists an arm $k$ such that $p_{ik} < 0.5$, but $u_{ik} \geq 0.5$.
   - By Lemma 9, for all time steps after $C(\delta)$, we have $l_{ik} \leq p_{ik} < 0.5$, and so in the second round we have $u_{ki} > 0.5 = u_{ii}$, and so $a_i$ could be not chosen as the challenger to itself.

   (b) $i \neq j$: In the case that $a_i$ is not a Copeland winner and $a_j$ is different from $a_i$, we distinguish between the following two cases, where $\Delta_i^*$ is defined as in Tables 2 and 3:

     i. $p_{ij} \leq 0.5 - \Delta_i^*$: In this case, the definition of $\Delta_i^*$ reduces to $\Delta_{ij}$. Now, since when choosing the challenger, CCB eliminates from consideration any arm $a_j$ that has $l_{ji} > 0.5$, the last time-step $\tau_{ij}$ after $C(\delta)$ when $a_j$ was chosen as the challenger for $a_i$, we must've had $u_{ij}(\tau_{ij}) := 1 - l_{ji}(\tau_{ij}) \geq 0.5$. On the other hand, Lemma 9 implies that

we must also have $l_{ij}(\tau_{ij}) \le p_{ij}$, and therefore, we have $u_{ij}(\tau_{ij}) - l_{ij}(\tau_{ij}) \ge \Delta_{ij}$; so, doing the same calculation as in part 1 of this proof, we have

$$u_{ij}(\tau_{ij}) - l_{ij}(\tau_{ij}) := 2\sqrt{\frac{\alpha \ln \tau_{ij}}{N_{ij}(\tau_{ij})}} \ge \Delta_{ij}$$

$$\therefore \quad 2\sqrt{\frac{\alpha \ln \tau_{ij}}{N_{ij}^{\delta}(\tau_{ij})}} \ge \Delta_{ij} \quad \because N_{ij}^{\delta}(\tau_{ij}) \le N_{ij}(\tau_{ij})$$

$$\therefore \quad 2\sqrt{\frac{\alpha \ln T}{N_{ij}^{\delta}(\tau_{ij})}} \ge \Delta_{ij} \quad \because \tau_{ij} \le T$$

$$\therefore \quad N_{ij}^{\delta}(\tau_{ij}) \le \frac{4\alpha \ln T}{\Delta_{ij}^2},$$

ii. $p_{ij} > 0.5 - \Delta_i^*$: Repeating the above argument about $u_{ij}(\tau_{ij})$, we can deduce that $u_{ij}(\tau_{ij}) \ge 0.5$ must hold. On the other hand, Lemma 9 states that with probability $1 - \delta$ we have $u_{ij}(\tau_{ij}) \ge p_{ij}$. Putting these two together we get

$$u_{ij}(\tau_{ij}) \ge \max\{0.5, p_{ij}\}. \tag{4}$$

On the other hand, we will show next that with probability $1 - \delta$, we have $l_{ij}(\tau_{ij}) \le 0.5 - \Delta_i^*$; this is a consequence of the following facts:

- Since $a_i$ was chosen as the optimistic Copeland winner, we can deduce that $a_i$ had no more that $L_C$ optimistic losses.
- Let $a_{k_1}, \ldots, a_{k_l}$ be the $l \le L_C$ arms to which $a_i$ lost optimistically during time-step $\tau_{ij}$. Then, the smallest $p_{ik}$ with $k \notin \{k_1, \ldots, k_l\}$, must be less than to equal to the $\{L_C + 1\}^{th}$ smallest element in the set $\{p_{ik} \mid k = 1, \ldots, K\}$.
- This, in turn, is equal to the $\{L_C + 1\}^{th}$ smallest element in the set $\{p_{ik} | p_{ik} < 0.5\}$ (since this latter set of numbers are the smallest ones in the former set). But, this is equal to $0.5 - \Delta_i^*$ by definition.

So, we have the desired bound on $l_{ij}(\tau_{ij})$ and combining this with Inequality (4), we have

$$u_{ij}(\tau_{ij}) - l_{ij}(\tau_{ij}) \ge \max\{0, p_{ij} - 0.5\} + \Delta_i^* = \Delta_{ij}^*,$$

where the last equality follows directly from the definition of $\Delta_{ij}^*$ and the fact that $p_{ij} > 0.5 - \Delta_i^*$. Now, repeating the same calculations as before, we can conclude that with probability $1 - \delta$, we have

$$N_{ij}^{\delta}(\tau_{ij}) \le \frac{4\alpha \ln T}{\left(\Delta_{ij}^*\right)^2}.$$

A pictorial depiction of the various steps in this part of the proof can be found in Figure 8. □

Figure 8: This figure illustrates the definition of the quantities $\Delta_i^*$ and $\Delta_{ij}^*$ in the case that arm $a_i$ is not a Copeland winner, as well as the idea behind Case 2 in the proof of Proposition 11. The bottom row of plots in the figure corresponds to the confidence intervals around probabilities $p_{ij}$ (depicted using the blue dots) for $j = 1, \ldots, K$, while the top row corresponds to those for probabilities $p_{1j}$, where $a_1$ is by assumption one of the Copeland winners (although we could use any other Copeland winner instead).

The two boxes in the top row with red intervals represent arms to which $a_1$ loses (i.e. $p_{1j} < 0.5$), the number of which happens to be 2 in this example, which means that $L_C = 2$. Now, by Definition 10.3, $i^*$ is the index with the index $j$ with the $(L_C + 1)^{th}$ (in this case $3^{rd}$) lowest $p_{ij}$, and since the three lowest $p_{ij}$ in this example are $p_{iK}, p_{iC}$ and $p_{ii^*}$, this means that the column labeled as $a_{i^*}$ is indeed labeled correctly. Given this, Definition 10.4 tells us that $\Delta_i^*$ is the size of the gap shown in the block corresponding to pair $(a_i, a_{i^*})$.

Moreover, by Definition 10.5, the gap $\Delta_{ij}^*$ is defined using one of the following three cases: (1) if we have $p_{ij} < p_{ii^*}$ (as with the ones with red confidence intervals in the bottom row of plots), then we get $\Delta_{ij}^* := \Delta_{ij} = 0.5 - p_{ij}$; (2) if we have $p_{ii^*} < p_{ij} \leq 0.5$ (as in the plots in the $2^{nd}$, $3^{rd}$ and $7^{th}$ column of the bottom row), then we get $\Delta_{ij}^* := \Delta_i^*$; (3) if we have $0.5 < p_{ij}$ (as in the $1^{st}$ and $6^{th}$ column in the bottom row), then we get $\Delta_{ij}^* := \Delta_{ij} + \Delta_i^*$.

The reasoning behind this trichotomy is as follows: in the case of arms $a_j$ in group (1), they are not going to be chosen to be played against $a_i$ as soon as top of the interval goes below 0.5, and by Lemma 9, we know that the bottom of the interval will be below $p_{ij}$. In the case of the arms in groups (2) and (3), the bottom of their interval needs to be below $p_{ii^*}$ because otherwise that would mean that neither arm $a_{i^*}$ nor arms in group (1) were eligible to be included in the $\arg\max$ expression in Line 13 of Algorithm 1, which can only happen if we have $u_{ij} < 0.5$ for $j = i^*$ as well as the arms in group (1), from which we can deduce that the optimistic Copeland score of $a_i$ must have been lower than $K - 1 - L_C$, and so $a_i$ could not have been chosen as an optimistic Copeland winner. Using the same argument, we can also see that the tops of the confidence intervals corresponding to arms in group (2) must be above 0.5, or else it would be impossible for $a_i$ to be chosen as an optimistic Copeland winner. Moreover, by Lemma 9, the intervals of the arms $a_j$ in group (3) must contain $p_{ij}$.

# G  Proof of Lemma 14

Let us begin with the following direct corollary of Proposition 11:

**Corollary 19.** *Given any $\delta > 0$, any $T > C(\delta)$ and any sub-interval of length $\widehat{N}^\delta(T) := \sum_{i \neq j} \widehat{N}^\delta_{ij}(T) + 1$, with probability $1 - \delta$, there is at least one time-step when there exists $c \leq C$ such that*

$$\underline{\mathrm{Cpld}}(a_c) = \mathrm{Cpld}(a_c) = \overline{\mathrm{Cpld}(a_c)}$$
$$\geq \overline{\mathrm{Cpld}(a_j)} \ \forall \, j, \qquad (5)$$

*Proof.* According to Proposition 11, with probability $1 - \delta$, there are at most $\sum_{i \neq j} \widehat{N}^\delta_{ij}(T)$ time-steps between $C(\delta)$ and $T$ when Algorithm 1 did not compare a Copeland winner against itself: i.e. $c$ and $d$ in Algorithm 1 did not satisfy $c = d \leq C$.

In other words, during this time-period, in any sub-interval of length $\widehat{N}^\delta(T) := \sum_{i \neq j} \widehat{N}^\delta_{ij}(T) + 1$, there is at least one time-step when a Copeland winner was compared against itself. During this time-step, we must have had

$$\underline{\mathrm{Cpld}}(a_c) = \mathrm{Cpld}(a_c) = \overline{\mathrm{Cpld}(a_c)}$$
$$\geq \overline{\mathrm{Cpld}(a_j)} \ \forall \, j,$$

where the first two equalities are due to the fact that in order for Algorithm 1 to set $c = d$, we must have $0.5 \notin [l_{cj}, u_{cj}]$ for each $j \neq c$, or else $a_c$ would not be played against itself; on the other hand, the last inequality is due to the fact that $a_c$ was chosen as an optimistic Copeland winner by Line 8 of Algorithm 1, so its optimistic Copeland score must have been greater than or equal to the optimistic Copeland score of the rest of the arms. $\square$

**Lemma 20.** *If there exists an arm $a_i$ with $i > C$ such that $\mathcal{B}^i_{C(\delta/2)}$ contains an arm $a_j$ that loses to $a_i$ (i.e. $p_{ij} > 0.5$) or such that $\mathcal{B}^i_{C(\delta/2)}$ contains fewer than $L_C + 1$ arms, then the probability that by time-step $T_0$ the sets $\mathcal{B}^i_t$ and $\mathcal{B}_t$ are not reset by Line 9.A of Algorithm 1 is less than $\delta/6$, where we define*

$$T_0 := C(\delta/2) + \widehat{N}^{\delta/2}(T_\delta)$$
$$+ \frac{32\alpha K(L_C + 1)\ln T_\delta}{\Delta^2_{\min}}$$
$$+ 8K^2(L_C + 1)^2 \ln \frac{6K^2}{\delta}.$$

*Proof.* By Line 9.A of Algorithm 1, as soon as we have $l_{ij} > 0.5$, the set $\mathcal{B}^i_t$ will be emptied. In what follows, we will show that the probability that the number of time-steps before we have $l_{ij} > 0.5$ is greater than

$$\Delta T := \widehat{N}^{\delta/2}(T_\delta) + N$$

with

$$N := \frac{32\alpha K(L_C + 1)\ln T_\delta}{\Delta^2_{\min}} + 8K^2(L_C + 1)^2 \ln \frac{6K^2}{\delta}$$

is bounded by $\delta/6K^2$. This is done using the amount of exploration infused by Line 10 of Algorithm 1. To begin, let us note that by Corollary 19, there is a time-step before $T_0 := C(\delta/2) + \widehat{N}^{\delta/2}(T_\delta)$ when the condition of Line 9.C of Algorithm 1 is satisfied for some Copeland winner. At this point, if $\mathcal{B}^i_t$ contains fewer than $L_C + 1$ elements, then it will be emptied; furthermore, for all $k > C$, the sets $B^k_{T_0}$ will have at most $L_C + 1$ elements and so the set

$$\mathcal{S}_t := \{(k, \ell) | a_\ell \in \mathcal{B}^k_t \text{ and } 0.5 \in [l_{k\ell}, u_{k\ell}]\}$$

contains at most $K(L_C + 1)$ elements for all $t \geq T_0$. Moreover, if at time-step $T_1 := C(\delta/2) + \Delta T$ we have $a_j \in \mathcal{B}^i_{T_1}$, then we can conclude that $(i, j) \in \mathcal{S}_t$ for all $t \in [C(\delta/2), T_1]$, since, if at any

time after $C(\delta/2)$ arm $a_j$ were to be removed from $\mathcal{B}_t^i$, it will never be added back because that can only happen through Line 9.B of Algorithm 1 and by Lemma 9 and the assumption of the lemma we have $u_{ij} > p_{ij} > 0.5$.

What we can conclude from the observations in the last paragraph is that if at time-step $T_1$ we still have $a_j \in \mathcal{B}_{T_1}^i$, then there are $\Delta T$ time-steps during which the probability of comparing arms $a_i$ and $a_j$ was at least $\frac{1}{4K(L_C+1)}$ and yet no more than $\frac{4\alpha \ln T_\delta}{\Delta_{ij}^2}$ comparisons took place, since otherwise, we would have $l_{ij} > 0.5$ at some point before $T_1$. Now, let $B_n^{ij}$ denote the indicator random variable that is equal to 1 if arms $a_i$ and $a_j$ were chosen to be played against each other by Line 10 of Algorithm 1 during time-step $T_1 + n$. Also, let $X_1, \ldots, X_N$ be iid Bernoulli random variables with mean $\frac{1}{4K(L_C+1)}$. Since $B_n^{ij}$ and $X_n$ are Bernoulli and we have $\mathbb{E}\left[B_n^{ij}\right] \leq \mathbb{E}[X_n]$ for each $n$, then we can conclude that

$$P\left(\sum_{n=1}^N B_n^{ij} < s\right) \leq P\left(\sum_{n=1}^N X_n < s\right) \quad \text{for all } s.$$

On the other hand, we can use the Hoeffding bound to show that the right hand side of the above inequality is smaller than $\delta/6$ if we set $s = \frac{4\alpha \ln T_\delta}{\Delta_{ij}^2}$:

$$P\left(\sum_{n=1}^N X_n < \frac{4\alpha \ln T_\delta}{\Delta_{ij}^2}\right) \leq P\left(\sum_{n=1}^N X_n < \frac{4\alpha \ln T_\delta}{\Delta_{\min}^2}\right)$$

$$= P\left(\sum_{n=1}^N X_n < \frac{N}{4K(L_C+1)} - a\right) \leq e^{-\frac{2a^2}{N}}$$

$$\text{with } a := -\frac{4\alpha \ln T_\delta}{\Delta_{\min}^2} + \frac{N}{4K(L_C+1)}$$

$$= e^{-\frac{32\alpha^2 \ln^2 T_\delta}{\Delta_{\min}^4 N} + \frac{4\alpha \ln T_\delta}{K(L_C+1)\Delta_{\min}^2} - \frac{N}{8K^2(L_C+1)^2}}$$

$$\leq e^{\frac{4\alpha \ln T_\delta}{K(L_C+1)\Delta_{\min}^2} - \frac{N}{8K^2(L_C+1)^2}}$$

$$= e^{-\ln 6K^2/\delta} = \delta/6K^2.$$

Now, if we take a union bound over all pairs of arms $a_i$ and $a_j$ satisfying the condition stated at the beginning of this scenario, we get that with probability $\delta/6$ by time-step $C(\delta/2) + \Delta T$ all such erroneous hypotheses are reset by Line 9.A of Algorithm 1, emptying the sets $\mathcal{B}_t^i$. $\square$

**Lemma 21.** *Let $t_1 \in [C(\delta/2), T_\delta)$ be such that for all $i, j$ satisfying $a_j \in \mathcal{B}_{t_1}^i$ we have $p_{ij} < 0.5$. Then, the following two statements hold with probability $1 - 5\delta/6$:*

1. *If the set $\mathcal{B}_{t_1}$ in Algorithm 1 contains at least one Copeland winner, then if we set $t_2 = t_1 + n_{\max}$, where*

$$n_{\max} := 2K \max_{i>C} \widehat{N}_i^{\delta/2}(T_\delta) + \frac{K^2 \ln(6K/\delta)}{2},$$

   *then $\mathcal{B}_{t_2}$ is non-empty and contains no non-Copeland winners, i.e. for all $a_i \in \mathcal{B}_{t_2}$ we have $i \leq C$.*

2. *If the set $\mathcal{B}_{t_1}$ in Algorithm 1 contains no Copeland winners, i.e. for all $a_i \in \mathcal{B}_{t_1}$, we have $i > C$, then within $n_{\max}$ time-steps the set $\mathcal{B}_t$ will be emptied by Line 9.B of Algorithm 1.*

*Therefore, with probability $1 - 5\delta/6$, by time $t_1 + 2n_{\max}$ all non-Copeland winners (i.e. arms $a_i$ with $i > C$) are eliminated from $\mathcal{B}_t$.*

*Proof.* We will consider the two cases in the following, conditioning on the conclusions of Lemma 9, Proposition 11 and Corollary 19, all simultaneously holding with $1 - \delta/2$:

1. $\mathcal{B}_{t_1}$ **contains** a Copeland winner (i.e. $a_c \in \mathcal{B}_{t_1}$ for some $c \leq C$): in this case, by Lemma 9, we know that the Copeland winner will forever remain in the set $\mathcal{B}_t$ because

$$\overline{\mathrm{Cpld}}(a_c) \geq \max_j \mathrm{Cpld}(a_j) \geq \max_j \underline{\mathrm{Cpld}}(a_j),$$

   then $\mathcal{B}_{t_2}$ will indeed be empty. Moreover, in what follows, we will show that the probability that any non-Copeland winner in $\mathcal{B}_t$ is not eliminated by time $t_2$ is less than $\delta/6$. Let us assume by way of contradiction that there exists an arm $a_b$ with $b > C$ such that $a_b$ is in $\mathcal{B}_{t_2}$: we will show that the probability of this happening is less than $\delta/6K$, and so, taking a union bound over non-Copeland winning arms, the probability that any non-Copeland winner is in $\mathcal{B}_{t_2}$ is seen to be smaller than $\delta/6$.

   Now, to see that the probability of $a_b$ being in the set $\mathcal{B}_{t_2}$ is small, note that the fact that $a_b$ being in $\mathcal{B}_{t_2}$ implies that $a_b$ was in the set $\mathcal{B}_t$ for the entirety of the time interval $[C(\delta/2), t_2]$ as we will show in the following. If $a_b$ is eliminated from $\mathcal{B}_t$ at some point between $t_1$ and $t_2$, it will not get added back into $\mathcal{B}_t$ because that can only take place if the set $\mathcal{B}_t$ is reset at some point and there are only two ways for that to happen:

   (a) By Line 9.A of Algorithm 1 in the case that for some pair $(i,j)$ with $a_j \in \mathcal{B}_t^i$ we have $l_{ij} > 0.5$; however, this is ruled out by our assumption that at time $t_1$ we have $p_{ij} < 0.5$ and by Lemma 9, which stipulates that we have $l_{ij} \leq p_{ij} < 0.5$.
   (b) By Line 9.B of Algorithm 1 in the case that all arms are eliminated from $\mathcal{B}_t$, but this cannot happen by the fact mentioned above that $a_c$ will not not be removed from $\mathcal{B}_t$.

   So, as mentioned above, we indeed have that at each time-step between $t_1$ and $t_2$, the set $\mathcal{B}_t$ contains $a_b$. Next, we will show that the probability of this happening is less than $\delta/6K$. To do so, let us denote by $\mathcal{S}_b$ the time-steps when arm $a_b$ was in the set of optimistic Copeland winners, i.e.

$$\mathcal{S}_b := \left\{ t \in (t_1, t_2] \,\middle|\, a_b \in \mathcal{C}_t \right\}.$$

   We can use Corollary 19 above with $T = T_\delta$ to show that the size of the set $\mathcal{S}_b$ (which we denote by $|\mathcal{S}_b|$) is bounded from below by $t_2 - t_1 - \sum_{i \neq j} \widehat{N}_{ij}^{\delta/2}(T_\delta)$: this is because whenever any Copeland winner $a_c$ is played against itself, Equation (5) holds, and so if we were to have $a_b \notin \mathcal{C}_t$ during that time-step $a_b$ would have had to get eliminated from $\mathcal{B}_t$ because $a_b$ not being an optimistic Copeland winner would imply that

$$\overline{\mathrm{Cpld}}(a_b) < \underline{\mathrm{Cpld}}(a_c) = \overline{\mathrm{Cpld}}(a_c).$$

   But, we know from facts (a) and (b) above that $a_b$ remains in $\mathcal{B}_t$ for all $t \in (t_1, t_2]$. Therefore, as claimed, we have

$$|\mathcal{S}_b| \geq t_2 - t_1 - \sum_{i \neq j} N_{ij}^{\delta/2}(T_\delta) \geq 2K\widehat{N}_b^{\delta/2}(T_\delta) + \frac{K^2 \ln(6K/\delta)}{2} =: n_b, \tag{6}$$

   where the last inequality is due to the definition of $n_{\max} := t_2 - t_1$. On the other hand, Proposition 11 tells us that the number of time-steps between $t_1$ and $t_2$ when $a_b$ could have been chosen as an optimistic Copeland winner is bounded as

$$N_b^{\delta/2}(T_\delta) \leq \widehat{N}_b^{\delta/2}(T_\delta). \tag{7}$$

   Furthermore, given the fact that during each time-step $t \in \mathcal{S}_b$ we have $a_b \in \mathcal{B}_t \cap \mathcal{C}_t$, the probability of $a_b$ being chosen as an optimistic Copeland winner is at least $1/K$ because of the sampling procedure in Lines 14-17 of Algorithm 1. However, this is considerably higher than the ratio obtained by dividing the right-hand sides of Inequality (7) by that of Inequality (6). We will make this more precise in the following: for each $t \in \mathcal{S}_b$, denote by $\mu_t^b$ the probability that arm $a_b$ would be chosen as the optimistic Copeland winner by Algorithm 1, and let $X_t^b$ be the Bernoulli random variable that returns 1 when arm $a_b$ is chosen as the optimistic Copeland winner

or 0 otherwise. As pointed out above, we have that $\mu_t^b \geq \frac{1}{K}$ for all $t \in \mathcal{S}_b$, which, together with the fact that $|\mathcal{S}_b| \geq n_b$, implies that the random variable $X^b := \sum_{t \in \mathcal{S}_b} X_t^b$ satisfies

$$P(X_b < x) \leq P(Binom(n_b, 1/K) < x). \tag{8}$$

This is both because the Bernoulli summands of $X_b$ have higher means than the Bernoulli summands of $Binom(n_b, 1/K)$ and because $X_b$ is the sum of a larger number of Bernoulli variables, so $X_b$ has more mass away from 0 than does $Binom(n_b, 1/K)$. So, we can bound the right-hand side of Inequality (8) by $\delta/6K$ with $x = \widehat{N}_b^{\delta/2}(T_\delta)$ to get our desired result. But, this is a simple consequence of the Hoeffding bound, a more general form of which is quoted in Section D. More precisely, we have

$$P\left(Binom(n_b, 1/K) < \widehat{N}_b^{\delta/2}(T_\delta)\right) = P\left(Binom(n_b, 1/K) < \frac{n_b}{K} - a\right)$$
$$\text{with } a := \frac{n_b}{K} - \widehat{N}_b^{\delta/2}(T_\delta)$$
$$< e^{-2a^2/n_b} = e^{\frac{-2\left(\frac{n_b}{K} - \widehat{N}_b^{\delta/2}(T_\delta)\right)^2}{n_b}}$$
$$= e^{-2n_b/K^2 + 4\widehat{N}_b^{\delta/2}(T_\delta)/K - 2\widehat{N}_b^{\delta/2}(T_\delta)^2/n_b}$$
$$\leq e^{-2n_b/K^2 + 4\widehat{N}_b^{\delta/2}(T_\delta)/K} = e^{-\ln(6K/\delta)} = \delta/6K$$

Using the union bound over the non-Copeland winning arms that were in $\mathcal{B}_{t_1}$, of whom there is at most $K - 1$, we can conclude that with probability $\delta/6$ they are all eliminated from $\mathcal{B}_{t_2}$.

2. $\mathcal{B}_{t_1}$ **does not contain** any Copeland winners: in this case, we can use the exact same argument as above to conclude that the probability that the set $\mathcal{B}_t$ is non-empty for all $t \in (t_1, t_2]$ is less than $\delta/6$ because as before the probability that each arm $a_b \in \mathcal{B}_{t_1}$ is not eliminated within $n_b$ time-steps is smaller than $\delta/6K$. $\qquad\square$

Let us now state the following consequence of the previous lemmas:

**Lemma 14.** *Given $\delta > 0$, the following fact holds with probability $1 - \delta$: for each $i > C$, the set $\mathcal{B}_{T_\delta}^i$ contains exactly $L_C + 1$ elements with each element $a_j$ satisfying $p_{ij} < 0.5$. Moreover, for all $t \in [T_\delta, T]$, we have $\mathcal{B}_t^i = \mathcal{B}_{T_\delta}^i$.*

*Proof.* In the remainder of the proof, we will condition on the high probability event that the conclusions of Lemma 9, Corollary 19, Lemma 20 and Lemma 21 all hold simultaneously with probability $1 - \delta$.

Combining Lemma 21, we can conclude that by time-step $T_1 := T_0 + 2n_{\max}$ all non-Copeland winners are removed from $\mathcal{B}_{T_1}$, which also means by Line 9.B of Algorithm 1 that the corresponding sets $\mathcal{B}_{T_1}^i$, with $i > C$ are non-empty, and Lemma 20 tells us that these sets have at least $L_C + 1$ elements $a_j$ each of which beats $a_i$ (i.e. $p_{ij} < 0.5$).

Now, applying Corollary 19, we know that within $\widehat{N}^{\delta/2}(T_\delta)$ time-steps, Line 9.C of Algorithm 1 will be executed, at which point we will have $\overline{L}_C = L_C$ and so $\mathcal{B}_t^i$ will be reduced to $L_C + 1$ elements. Moreover, by Lemma 9, for all $t > T_1$ and $a_j \in \mathcal{B}_t^i$ we have $l_{ij} \leq p_{ij} < 0.5$ and so $\mathcal{B}_t^i$ will not be emptied by any of the provisions in Line 9 of Algorithm 1.

Now, since by definition we have $T^\delta \geq T_1 + \widehat{N}^{\delta/2}(T_\delta)$, we have the desired result. $\qquad\square$

# H  Proof of Lemma 15

**Lemma 15** *Given a Copeland bandit problem satisfying Assumption* **A** *and any* $\delta > 0$, *with probability* $1 - \delta$ *the following statement holds: the number of time-steps between* $T_{\delta/2}$ *and* $T$ *when each non-Copeland winning arm* $a_i$ *can be chosen as optimistic Copeland winners (i.e. time-steps when arm* $a_c$ *in Algorithm 1 satisfies* $c = i > C$*) is bounded by*

$$\widehat{N}^i := 2\widehat{N}^i_{\mathcal{B}} + 2\sqrt{\widehat{N}^i_{\mathcal{B}}} \ln \frac{2K}{\delta},$$

*where*

$$\widehat{N}^i_{\mathcal{B}} := \sum_{j \in \mathcal{B}^i_{T_{\delta/2}}} \widehat{N}^{\delta/4}_{ij}(T).$$

*Proof.* The idea of the argument is outlined in the following sequence of facts:

1. By Lemma 14, we know that with probability $1 - \delta/2$, for each $i > C$ and all times $t > T_{\delta/2}$ the sets $\mathcal{B}^i_t$ will consist of exactly $L_C + 1$ arms that beat the arm $a_i$, and that $\mathcal{B}^i_t = \mathcal{B}^i_{T_{\delta/2}}$.

2. Moreover, if at time $t > T_{\delta/2} > C(\delta/4)$, Algorithm 1 chooses a non-Copeland winner as an optimistic Copeland winner (i.e. $i > C$), then with probability $1 - \delta/4$ we know that

$$\overline{\mathrm{Cpld}}(a_i) \geq \overline{\mathrm{Cpld}}(a_1) \geq \mathrm{Cpld}(a_1) = K - 1 - L_C.$$

3. This means that there could be at most $L_C$ arms $a_j$ that optimistically lose to $a_i$ (i.e. $u_{ij} < 0.5$) and so at least one arm $a_b \in \mathcal{B}^i_t$ does satisfy $u_{ib} \geq 0.5$

4. This, in turn, means that in Line 13 of Algorithm 1 with probability $0.5$ the arm $a_d$ will be chosen from $\mathcal{B}^i_t$.

5. By Proposition 11, we know that with probability $1 - \delta/4$, in the time interval $[T_{\delta/2}, T]$ each arm $a_j \in \mathcal{B}^i_{T_{\delta/2}}$ can be compared against $a_i$ at most $\widehat{N}^{\delta/4}_{ij}(T)$ many times.

Given that by Fact 3 above we need at least one arm $a_j \in \mathcal{B}^i_t$ to satisfy $u_{ij} \geq 0.5$ for Algorithm 1 to set $(c, d) = (i, j)$, and that by Fact 4 arms from $\mathcal{B}^i_t$ have a higher probability of being chosen to be compared against $a_i$, this means that arm $a_i$ will be chosen as optimistic Copeland winner roughly twice as many times we had $(c, d) = (i, j)$ for some $j \in \mathcal{B}^i_{T_{\delta/2}}$. A high probability version of the claim in the last sentence together with Fact 5 would give us the bound on regret claimed by the theorem. In the remainder of this proof, we will show that indeed the number of times we have $c = i$ is unlikely to be too many times higher than twice the number of times we get $(c, d) = (i, j)$, where $j \in \mathcal{B}^i_{T_{\delta/2}}$. To do so, we will introduce the following notation:

$N^i$: the number of time-steps between $T_{\delta/2}$ and $T$ when arm $a_i$ was chosen as optimistic Copeland winner.

$B^i_n$: the indicator random variable that is equal to 1 if Line 13 in Algorithm 1 decided to choose arm $a_d$ only from the set $B^i_{t_n}$ and zero otherwise, where $t_n$ is the $n^{th}$ time-step after $T_{\delta/2}$ when arm $a_i$ was chosen as optimistic Copeland winner. Note that $B^i$ is simply a Bernoulli random variable mean 0.5.

$N^i_{\mathcal{B}}$: the number of time-steps between $T_\delta$ and $T$ when arm $a_i$ was chosen as optimistic Copeland winner and that Line 13 in Algorithm 1 chose to pick an arm from $\mathcal{B}^i_{T_{\delta/2}}$ to be played against $a_i$. Note that this definition implies that we have

$$N^i_{\mathcal{B}} = \sum_{n=1}^{N^i} B^i_n. \tag{9}$$

Moreover, by Fact 5 above, we know that with probability $1 - \delta/4$ we have

$$N^i_{\mathcal{B}} \leq \widehat{N}^i_{\mathcal{B}} := \sum_{j \in \mathcal{B}^i_{T_{\delta/2}}} \widehat{N}^{\delta/4}_{ij}(T). \tag{10}$$

Now, we will use the above high probability bound on $N_{\mathcal{B}}^i$ to put the following high probability bound on $N^i$: with probability $1 - \delta/2$ we have

$$N^i \leq \widehat{N}^i := 2\widehat{N}_{\mathcal{B}}^i + 2\sqrt{\widehat{N}_{\mathcal{B}}^i} \ln \frac{2K}{\delta}.$$

To do so, let us assume that the we have $N^i > \widehat{N}^i$ and consider the first $\widehat{N}^i$ time-steps after $T_{\delta/2}$ when arm $a_i$ was chosen as optimistic Copeland winner and note that by Equation (9) we have

$$\sum_{n=1}^{\widehat{N}^i} B_n^i \leq N_{\mathcal{B}}^i$$

and so by Inequality (10) with probability $1 - \delta/4$ the left-hand side of the last inequality is bounded by $\widehat{N}_{\mathcal{B}}^i$: let us denote this event with $\mathcal{E}$. On the other hand, if we apply the Hoeffding bound (cf. Appendix D) to the variables $B_1^i, \ldots, B_{\widehat{N}^i}^i$, we get

$$
\begin{aligned}
P\left(\mathcal{E} \wedge N^i > \widehat{N}^i\right) &\leq P\left(\sum_{n=1}^{\widehat{N}^i} B_n^i < \widehat{N}_{\mathcal{B}}^i\right) \\
&= P\left(\sum_{n=1}^{\widehat{N}^i} B_n^i < \widehat{N}^i/2 - \sqrt{\widehat{N}_{\mathcal{B}}^i} \ln \frac{2K}{\delta}\right) \\
&\leq e^{-\dfrac{2\widehat{N}_{\mathcal{B}}^i \left(\ln \frac{2K}{\delta}\right)^2}{2\widehat{N}_{\mathcal{B}}^i + 2\sqrt{\widehat{N}_{\mathcal{B}}^i} \ln \frac{2K}{\delta}}}
\end{aligned}
\tag{11}
$$

To simplify the last expression in the last chain of inequalities, let us use the notation $\alpha := \widehat{N}_{\mathcal{B}}^i$ and $\beta := \ln \frac{2K}{\delta}$. Given this notation, we claim that the following inequality holds if we have $\alpha \geq 4$ and $\beta \geq 2$ (which hold by the assumptions of the theorem):

$$\frac{\alpha\beta^2}{\alpha + \sqrt{\alpha}\beta} \geq \beta.\tag{12}$$

To see this, let us multiply both sides by the denominator of the left-hand side of the above inequality:

$$\alpha\beta^2 \geq \alpha\beta + \sqrt{\alpha}\beta.\tag{13}$$

To see why Inequality (13) holds, let us note that the restrictions imposed on $\alpha$ and $\beta$ imply the following pair of inequalities, whose sum is equivalent to Inequality (13):

$$
\begin{array}{rcl}
\alpha\beta^2 & \geq & 2\alpha\beta \\
+ \quad \alpha\beta^2 & \geq & 2\sqrt{\alpha}\beta^2 \\
\hline
= \quad 2\alpha\beta^2 & \geq & 2\alpha\beta + 2\sqrt{\alpha}\beta^2
\end{array}
$$

Now that we know that Inequality (12) holds, we can combine it with Inequality (11) to get

$$P\left(\mathcal{E} \wedge N^i > \widehat{N}^i\right) \leq e^{-\ln \frac{2K}{\delta}} = \frac{\delta}{2K}.$$

Taking a union over the non-Copeland winning arms, we get

$$P(\mathcal{E} \wedge \forall i > C, N^i > \widehat{N}^i) > 1 - \delta/2.$$

So, given the fact that we have $P(\mathcal{E}) < \delta/4$, we know that with probability $1 - \delta$ each non-Copeland winner is selected as optimistic Copeland winner between $T_{\delta/2}$ and $T$ no more than $\widehat{N}^i$ times. $\quad\square$

# I A Scalable Solution to the Copeland Bandit Problem

In this section, we prove Lemma 6, providing an analysis to the PAC solver of the Copeland winner identification algorithm.

To simplify the proof, we begin by solving a slightly easier variant of Lemma 6 where the queries are deterministic. Specifically, rather than having a query to the pair $(a_i, a_j)$ be an outcome of a Bernoulli r.v. with an expected value of $p_{ij}$, we assume that such a query simply yields the answer to whether $p_{ij} > 0.5$. Clearly, a solution can be obtained using $K(K-1)/2$ many queries but we aim for a solution with query complexity linear in $K$. In this section we prove the following.

**Lemma 22.** *Given $K$ arms and a parameter $\epsilon$, Algorithm 2 finds a $(1+\epsilon)$-approximate best arm with probability at least $1 - \delta$, by using at most*

$$\log(K/\delta) \cdot \mathcal{O}\left(K\log(K) + \min\left\{\frac{K}{\epsilon^2}, K^2(1 - \mathrm{cpld}(a_1))\right\}\right)$$

*many queries. In particular, when there is a Condorcet winner ($\mathrm{cpld}(a_1) = 1$) or more generally $\mathrm{cpld}(a_1) = 1 - \mathcal{O}(1/K)$, an exact solution can be found with probability at least $1 - \delta$ by using at most*

$$\mathcal{O}\left(K\log(K)\log(K/\delta)\right)$$

*many queries.*

The idea behind our algorithm is as follows. We provide an unbiased estimator of the normalized Copeland score of arm $a_i$ by picking an arm $a_j$ uniformly at random and querying the pair $(a_i, a_j)$. This method allows us to apply proof techniques for the classic MAB problem. These techniques provide a bound on the number of queries dependent on the gaps between the different Copeland scores. Our result is obtained by noticing that there cannot be too many arms with a large Copeland score; the formal statement is given later in Lemma 7. If the Copeland winner has a large Copeland score, i.e., $L_C$ is small, then only a small number of arms can be close to optimal. Hence, the main argument of the proof is that the majority of arms can be eliminated quickly and only a handful of arms must be queried many times.

As stated above, our algorithm uses as a black box Algorithm 4, an approximate-best-arm identification algorithm for the classical MAB setup. Recall that here, each arm $a_i$ has an associated reward $\mu_i$ and the objective is to identify an arm with the (approximately) largest reward. Without loss of geenrality, we assume that $\mu_1$ is the maximal reward. The following lemma provides an analysis of Algorithm 4 that is tight for the case where $\mu_1$ is close to 1. In this case, it is exactly the set of near optimal arms that will be queried many times hence it is important to take into consideration that the random variables associated with near optimal arms have a variance of roughly $1 - \mu_i$, which can be quite small. This translates to savings in the number of queries to arm $a_i$ by a factor of $1 - \mu_i$ compared to an algorithm that does not take the variances into account.

**Lemma 23.** *Algorithm 4 requires as input an error parameter $\epsilon$, failure probability $\delta$ and an oracle to $k$ Bernoulli distributions. It outputs, with probability at least $1 - \delta$, a $(1 + \epsilon)$-approximate best arm, that is an arm $a_i$ with corresponding expected reward of $\mu \geq 1 - (1 - \mu_1)(1 + \epsilon)$ with $\mu_1$ being the maximum expected value among arms. The expected number of queries made by the algorithm is upper bounded by*

$$\mathcal{O}\left(\sum_i \frac{(1 - \mu_i)\log(K/(\delta\Delta_i\epsilon))}{(\Delta_i^\epsilon)^2}\right),$$

*with $\Delta_i^\epsilon = \max\{\mu_1 - \mu_i, \epsilon(1 - \mu_1)\}$. Moreover, with probability at least $1 - \delta$, the number of times arm $i$ will be queried is at most*

$$\mathcal{O}\left(\frac{(1 - \mu_i)\log(K/(\delta\Delta_i\epsilon))}{(\Delta_i^\epsilon)^2}\right).$$

We prove Lemma 23 in Appendix J.

For convenience, we denote by $\mu_i$ the normalized Copeland score of arm $a_i$ and $\mu_1$ the maximal normalized Copeland score. To get an informative translation of the above expression to our setting,

let $A$ be the set of arms with normalized Copeland score in $(1 - 2(1 - \mu_1), \mu_1]$ and let $\bar{A}$ be the set of the other arms. In our setting, this query complexity of Algorithm 4 is upper bounded by

$$\mathcal{O}\left(\frac{2|A|\log(K/\delta)}{(1 - \mu_1)\epsilon^2} + \sum_{i \in \bar{A}} \frac{\log(K/\delta)(1 - \mu_i)}{(\mu_1 - \mu_i)^2}\right), \tag{14}$$

assuming[7] $\delta < (1 - \mu_1)\epsilon$.

It remains to provide an upper bound for the above expression given the structure of the normalized Copeland scores. In particular, we use the results of Lemma 7, repeated here for convenience.

**Lemma 7.** *Let $D \subset [K]$ be the set of arms for which $\mathrm{cpld}(a_i) \geq 1 - d/(K - 1)$, that is arms that are beaten by at most $d$ arms. Then $|D| \leq 2d + 1$.*

We bound the left summand in (14):

$$\frac{2|A|\log(K/\delta)}{(1 - \mu_1)\epsilon^2} \leq \frac{(4(1 - \mu_1)(K - 1) + 2)\log(K/\delta)}{(1 - \mu_1)\epsilon^2} = O\left(\frac{\log(K/\delta)K}{\epsilon^2}\right). \tag{15}$$

We now bound the right summand in (14). Let $i \in \bar{A}$. According to the definition of $\bar{A}$ it holds that $(1 - \mu_i) \leq 2(\mu_1 - \mu_i)$. Hence:

$$\sum_{i \in \bar{A}} \frac{\log(K/\delta)(1 - \mu_i)}{(\mu_1 - \mu_i)^2} \leq \sum_{i \in \bar{A}} \frac{4\log(K/\delta)}{1 - \mu_i}.$$

**Lemma 24.** *We have $\displaystyle\sum_{i:\, \mu_i < 1} \frac{1}{1 - \mu_i} = \mathcal{O}(K \log(K))$.*

*Proof.* Let $A_\tau$ be the set of arms for which $2^\tau \leq 1 - \mu_i < 2^{\tau+1}$. According to Lemma 7, we have that $|A_\tau| \leq 2^{\tau+2}(K - 1) + 1$. Other than that, since $1 \geq 1 - \mu_i \geq 1/(K - 1)$ for all $i > C$ we have that $A_\tau = \emptyset$ for any $\tau \leq -\log_2(K - 1) - 1$ and $\tau > 0$. It follows that:

$$\sum_{i > C} \frac{1}{1 - \mu_i} \leq \sum_{\ell=0}^{\lceil \log_2(K-1) \rceil} \frac{|A_{\ell - \log_2(K-1)}|}{2^{\ell - \log_2(K-1)}} \leq \sum_{\ell=0}^{\lceil \log_2(K-1) \rceil} \frac{2^{2+\ell} + 1}{2^{\ell - \log_2(K-1)}}$$

$$\leq (\lceil \log_2(K - 1) \rceil + 1) \cdot 5(K - 1). \qquad \square$$

From (14), (15) and Lemma 24, we conclude that the total number of queries is bounded by

$$\mathcal{O}\left(\log(K/\delta)\left(K\log(K) + \frac{K}{\epsilon^2}\right)\right).$$

In order to prove Lemma 22, it remains to analyze the case where $\epsilon$ is extremely small. Specifically, when $\epsilon^2(1 - \mu_1)$ takes a value smaller than $1/K$ then the algorithm becomes inefficient in the sense that it queries the same pair more than once. This can be avoided by taking the samples of $j$ when querying the score of arm $a_i$ to be uniformly random *without* replacement. The same arguments hold but are more complex as now the arm pulls are not i.i.d. Nevertheless, the required concentration bounds still hold. The resulting argument is that the number of queries is $\tilde{O}\left(\log(1/\delta)\left(K + \frac{K}{\bar{\epsilon}^2}\right)\right)$ with $\bar{\epsilon} = \max\{\epsilon, 1/\left(\sqrt{K(1 - \mu_1)}\right)\}$. Lemma 22 immediately follows.

We are now ready to analyze the stochastic setting.

*Proof of Lemma 6.* By querying arm $a_i$ we choose a random arm $j \neq i$ and in fact query the pair $(a_i, a_j)$ sufficiently many times in order to determine whether $p_{ij} > 0.5$ with probability at least $1 - \delta/K^2$. Standard concentration bounds show that achieving this requires querying the pair $(a_i, a_j)$

at most $\mathcal{O}\left(\log(K/(\Delta_{ij}\delta))\Delta_{ij}^{-2}\right)$ many times. It follows that a single query to arm $a_i$ in the deterministic case translates into an expected number of

$$\mathcal{O}\left(\log(KH_i/\delta)\frac{H_i}{K-1}\right) = \mathcal{O}\left(\frac{\log(KH_\infty/\delta)H_\infty}{K}\right)$$

many queries in the stochastic setting. The claim now follows from the bound on the expected number of queries given in Lemma 22. $\qquad\square$

## J   KL-based approximate best arm identification algorithm

Algorithm 4 solves an approximate best arm identification problem using confidence bounds based on Chernoff's inequality stated w.r.t the KL-divergence of two random variables. Recall that for two Bernoulli random variables with parameters $p, q$ the KL-divergence from $q$ to $p$ is defined as $d(p,q) = (1-p)\ln((1-p)/(1-q))+p\ln(p/q)$ with $0\ln(0) = 0$. The building block of Algorithm 4 is the well known Chernoff bound stating that for a Bernoulli random variable with expected value $q$, the probability of the average of $n$ i.i.d samples from it to be smaller (larger) than $p$, for $p < q$ ($p > q$), is bounded by $\exp(-nd(p,q))$.

---

**Algorithm 4** KL-best arm identification

---

**Input:** Access to oracle giving a noisy approximation of the reward of arm $i$ for $K$ arms, success probability $\delta > 0$, approximation parameter $\epsilon > 0$
1: **for all** $i \in [K]$ **do**
2:     $T = 1$
3:     $S_i \leftarrow \text{reward}(i)$
4:     $I_i \leftarrow [0, 1]$
5: **end for**
6: $B \leftarrow [K]$
7: $t \leftarrow 2$
8: **while** $\frac{1-\max_{i\in B}\min I_i}{1-\max_{i\in B}\max I_i} > (1+\epsilon)$ **do**
9:     For all $i \in B$, $S_i \leftarrow S_i + \text{reward}(i)$
10:    For all $i \in B$, let $I_i = \{q \in [0,1], \ t \cdot d(\frac{S_i}{t}, q) \leq \ln(4tK/\delta) + 2\ln\ln(t)\}$
11:    For all $i \in B$ for which there exist some $j \in B$ with $\max\{q \in I_i\} < \min\{q \in I_j\}$, remove $i$ from $B$.
12:    $t \leftarrow t + 1$
13: **end while**
**Return:** $\arg\max_{i\in B}\min I_i$.

---

*Proof of Lemma 23.* We use an immediate application of the Chernoff-Hoeffding bound

**Lemma 25.** *Fix $i \in [K]$. Let $E_t^i$ denote the event that at iteration $t$, $\mu_i \notin I_i$. We have that $\Pr[E_t^i] \leq 2\frac{\delta}{4tK} \cdot \frac{1}{\log(t)^2} \leq \frac{\delta}{2t\log(t)^2 K}$.*

Let $E$ denote the union, over all $t, i$ of events $E_t^i$. That is, $E$ denotes the event in which there exist some iteration $t$, and for some arm $a_i$ such that $\mu_i \notin I_i$. By the above lemma we get that

$$\Pr[E] \leq \sum_t \sum_i \Pr[E_t^i] \leq K\sum_{t=2}^{\infty} \frac{\delta}{2t\log(t)^2 K} \leq \delta$$

It follows that given that event $E$ did not happen, the algorithm will never eliminate the top arm and furthermore, will output an $(1+\epsilon)$-approximate best arm. We proceed to analyze the total number of pulls per arm, while having a separate analysis for $(1+\epsilon)$-approximate best arms and the other arms. We begin by stating an auxiliary lemma giving explicit bounds for the confidence regions.

**Lemma 26.** *Assume that event $E$ did not occur and let $\rho \geq 0$. For a sufficiently large universal constant $c$ we have for any $t \geq \frac{c\log(tK/\delta)(1-\mu_i)}{\rho^2}$ that $\max I_i < \mu_i + \rho$. Also, for $t \geq \frac{c\log(tK/\delta)(1-\mu_i+\rho/2)}{\rho^2}$ it holds that $\min I_i > \mu - \rho$.*

*Proof.* We consider the Taylor series associated with $f(x) = d(p + x, p)$. Since $f(0) = f'(0) = 0$ it holds that for any $x \leq 1 - p$ there exists some $|x'| \leq |x|$ with

$$f(x) = x^2 f''(x') = \frac{x^2}{(p + x')(1 - p - x')} \leq \frac{2x^2}{1 - p}$$

To prove that $\max I_i < \mu_i + \rho$ we apply the above observation for $\rho \leq 1 - \mu_i$ (otherwise $\mu_i + \rho > 1$ and the claim is trivial) and reach the conclusion that for sufficiently large universal constant $c$ it holds that

$$t \cdot d(\mu_i + \rho/2, \mu_i) > \log(tK/\delta) + 2 \log \log(tK/\delta)$$
$$t \cdot d(\mu_i + \rho/2, \mu_i + \rho) > \log(tK/\delta) + 2 \log \log(tK/\delta)$$

The first inequality dictates that $S_i/t \leq \mu_i + \rho/2$. The second inequality dictates that $t \cdot d(S_i/t, \mu_i + \rho) \geq d(\mu_i + \rho/2, \mu_i + \rho)$ is too large in order for $\mu_i + \rho$ to be an element of $I_i$.

The bound for $\min I_i$ is analogous. Since now we have $t \geq \frac{c \log(tK/\delta)(1 - \mu_i + \rho/2)}{\rho^2}$, it holds that

$$t \cdot d(\mu_i - \rho/2, \mu_i) > \log(tK/\delta) + 2 \log \log(tK/\delta)$$

$$t \cdot d(\mu_i - \rho/2, \mu_i - \rho) > \log(tK/\delta) + 2 \log \log(tK/\delta)$$

This means that first, $S_i/t \geq \mu_i - \rho/2$ and second, that $t \cdot d(S_i/t, \mu_i - \rho) \geq d(\mu_i - \rho/2, \mu_i - \rho)$ is too large in order for $\mu_i - \rho$ to be an element of $I_i$. $\square$

**Lemma 27.** *Let $i$ be a suboptimal arm, meaning one where $\mu_i \leq 1 - (1 - \mu_1)(1 + \epsilon)$. Denote by $\Delta_i$ its gap $\mu_1 - \mu_i$. If event $E$ does not occur then $i$ is queried at most $O\left(\frac{\log\left(\frac{K}{\delta \Delta_i}\right) v_i}{(\Delta_i)^2}\right)$ many times, where $v_i = 1 - \mu_i$*

*Proof.* We first notice that as we are assuming that event $E$ did not happen, it must be the case that arm 1 is never eliminated from $B$. Consider an iteration $t$ such that

$$t \geq \frac{c \log(tK/\delta) v_i}{(\Delta_i)^2} \tag{16}$$

for a sufficiently large $c$, then according to Lemma 26 it holds that $\max I_i < \mu_i + \Delta_i/2$. Now, since $v_i = 1 - \mu_i \geq 1 - \mu_1 + \Delta_i/2$ we have that for the same $t$ it must be the case that $\min I_1 > \mu_1 - \Delta_i/2$. It follows that $\min I_1 > \max I_i$ and arm $a_i$ is eliminated at round $t$. $\square$

**Lemma 28.** *Assume $\epsilon \leq 1$. If event $E$ does not occur then for some sufficiently large universal constant $c$ it holds that when $t \geq \frac{c \log(tK/\delta)}{(1 - \mu_1)\epsilon^2}$ the algorithm terminates.*

*Proof.* Let $i$ be an arbitrary arm. Since

$$t \geq \frac{c \log(tK/\delta)}{(1 - \mu_1)\epsilon^2} = \frac{c \log(tK/\delta)(1 - \mu_i)}{(1 - \mu_1)(1 - \mu_i)\epsilon^2}$$

we get, according to Lemma 26 that

$$\max I_i \leq \mu_i + \frac{\epsilon}{3}\sqrt{(1 - \mu_i)(1 - \mu_1)}$$

In order to bound $\sqrt{(1 - \mu_i)(1 - \mu_1)}$ we consider the function $f(x) = \sqrt{v(v + x)}$. Notice that $f(0) = v$ and $f'(x) = \frac{v}{2\sqrt{v(v+x)}} \leq \frac{1}{2}$ for $x \geq 0$. It follows that for positive $x$, $\sqrt{v(v + x)} \leq v + x/2$, meaning that

$$\max I_i \leq \mu_i + \frac{\epsilon\left((1 - \mu_i) + \Delta_i/2\right)}{3} \leq \mu_1 + \frac{\epsilon(1 - \mu_1)}{3}$$

Now, since $\epsilon \leq 1$ we have

$$t \geq \frac{c \log(tK/\delta)(1 - \mu_1)}{(1 - \mu_1)^2 \epsilon^2} \geq \frac{(c/2) \log(tK/\delta)(1 - \mu_1 + \epsilon(1 - \mu_1))}{(1 - \mu_1)^2 \epsilon^2}$$

hence for sufficiently large $c$ we can apply Lemma 26 and obtain

$$\min I_1 \geq \mu_1 - \frac{\epsilon(1 - \mu_1)}{3}$$

It follows that assuming $\epsilon \leq 1$,

$$\min I_1 \geq 1 - \left(1 - \max_i I_i\right)(1 + \epsilon)$$

meaning that the algorithm will terminate at iteration $t$. ☐

This concludes the proof of Lemma 23 ☐

Table 2: List of notation used in this paper

| Symbol | Definition |
|---|---|
| $K$ | Number of arms |
| $[K]$ | The set $\{1, \dots, K\}$ |
| $a_1, \dots, a_K$ | Set of arms |
| $p_{ij}$ | Probability of arm $a_i$ beating arm $a_j$ |
| $\mathrm{Cpld}(a_i)$ | Copeland score: number of arms that $a_i$ beats, i.e. $|\{j \mid p_{ij} > 0.5\}|$ |
| $\mathrm{cpld}(a_i)$ | Normalized Copeland score: $\dfrac{\mathrm{Cpld}(a_i)}{K-1}$ |
| $C$ | Number of Copeland winners, i.e. arms $a_i$ with $\mathrm{Cpld}(a_i) \geq \mathrm{Cpld}(a_j)$ for all $j$ |
| $a_1, \dots, a_C$ | Copeland winner arms |
| $\alpha$ | UCB parameter of Algorithm 1 |
| $\delta$ | Probability of failure |
| $C(\delta)$ | $\left( \dfrac{(4\alpha - 1)K^2}{(2\alpha - 1)\delta} \right)^{\frac{1}{2\alpha - 1}}$ |
| $N_i(t)$ | Number of times arm $a_i$ was chosen as the optimistic Copeland winner until time $t$ |
| $N_i^\delta(t)$ | Number of times arm $a_i$ was chosen as the optimistic Copeland winner in the interval $(C(\delta), t]$ |
| $N_{ij}(t)$ | Total number of time-steps before $t$ when $a_i$ was compared against $a_j$ (notice that this definition is symmetric with respect to $i$ and $j$) |
| $N_{ij}^\delta(t)$ | Number of time-steps between times $C(\delta)$ and $t$ when $a_i$ was chosen as the optimistic Copeland winner and $a_j$ as the challenger (note that, unlike $N_{ij}(t)$, this definition is not symmetric with respect to $i$ and $j$) |
| $\tau_{ij}$ | The last time-step when $a_i$ was chosen as the optimistic Copeland winner and $a_j$ as the challenger (note that $\tau_{ij} \geq C(\delta)$ iff $N_{ij}^\delta(t) > 0$) |
| $w_{ij}(t)$ | Number of wins of $a_i$ over $a_j$ until time $t$ |
| $u_{ij}(t)$ | $\dfrac{w_{ij}(t)}{N_{ij}(t)} + \sqrt{\dfrac{\alpha \ln t}{N_{ij}(t)}}$ |
| $l_{ij}(t)$ | $1 - u_{ji}(t)$ |
| $\overline{\mathrm{Cpld}}(a_i)$ | $\#\left\{ k \mid u_{ik} \geq \frac{1}{2}, k \neq i \right\}$ |
| $\underline{\mathrm{Cpld}}(a_i)$ | $\#\left\{ k \mid l_{ik} \geq \frac{1}{2}, k \neq i \right\}$ |
| $\mathcal{C}_t$ | $\{ i \mid \overline{\mathrm{Cpld}}(a_i) = \max_j \overline{\mathrm{Cpld}}(a_j) \}$ |
| $\mathcal{L}_i$ | the set of arms to which $a_i$ loses, i.e. $a_j$ such that $p_{ij} < 0.5$ |
| $L_C$ | The largest number of losses that any Copeland winner has, i.e. $\max_{i=1}^{C} |\{j \mid p_{ij} < 0.5\}|$ |
| $\overline{L}_C$ | Algorithm 1's estimate of $L_C$ |
| $\mathcal{B}_t$ | The potentially best arms at time $t$, i.e. the set of arms that according to Algorithm 1 have some chance of being Copeland winners |
| $\mathcal{B}_t^i$ | The arms that at time $t$ have the best chance of beating arm $a_i$ (Cf. Line 12 in Algorithm 1) |
| $\Delta_{ij}$ | $|p_{ij} - 0.5|$ |
| $\Delta_{\min}$ | $\min\{\Delta_{ij} \mid \Delta_{ij} \neq 0\}$ |
| $i^*$ | the index of the $(L_C + 1)^{th}$ largest element in the set $\{\Delta_{ij} \mid p_{ij} < 0.5\}$ in the case that $i > C$ |
| $\Delta_i^*$ | $\begin{cases} \Delta_{ii^*} & \text{if } i > C \\ 0 & \text{otherwise} \end{cases}$ |

Table 3: List of notation used in this paper (Cont'd)

| Symbol | Definition |
|---|---|
| $\Delta_{ij}^*$ | $\begin{cases} \Delta_i^* + \Delta_{ij} & \text{if } p_{ij} \geq 0.5 \\ \max\{\Delta_i^*, \Delta_{ij}\} & \text{otherwise} \end{cases}$ <br><br> (See Figures 8 and 7 for a pictorial explanation.) |
| $\Delta_{\min}^*$ | $\min_{i>C} \Delta_i^*$ |
| $\widehat{N}_{ij}^\delta(T)$ | $\begin{cases} \frac{4\alpha \ln T}{\left(\Delta_{ij}^*\right)^2} & \text{if } i \neq j \\ 0 & \text{if } i = j \text{ and } i > C \end{cases}$ |
| $\widehat{N}_i^\delta(T)$ | $\sum_{j=1}^{K} \widehat{N}_{ij}^\delta(T)$ |
| $\widehat{N}^\delta(T)$ | $\sum_{i \neq j} \widehat{N}_{ij}^\delta(T) + 1$ |
| $T_\delta \geq$ | $C(\frac{\delta}{2}) + 8K^2(L_C+1)^2 \ln \frac{6K^2}{\delta} + K^2 \ln \frac{6K}{\delta}$ <br> $\qquad + \frac{32\alpha K(L_C+1)}{\Delta_{\min}^2} \ln T_\delta + \widehat{N}^{\delta/2}(T_\delta)$ <br> $\qquad + 4K \max_{i>C} \widehat{N}_i^{\delta/2}(T_\delta)$ <br><br> $T_\delta$ is the smallest integer satisfying the above inequality (Cf. Definition 12). |
| $T_0$ | $C(\delta/2) + \widehat{N}^{\delta/2}(T_\delta)$ <br> $\qquad + \frac{32\alpha K(L_C+1) \ln T_\delta}{\Delta_{\min}^2}$ <br> $\qquad + 8K^2(L_C+1)^2 \ln \frac{6K^2}{\delta}$ |
| $n_b$ | $2K \widehat{N}_b^{\delta/2}(\widehat{T}_\delta) + \frac{K^2 \ln(4K/\delta)}{2}$ |
| $Binom(n,p)$ | A "binomial" random variable obtained from the sum of $n$ independent Bernoulli random variables, each of which produces 1 with probability $p$ and 0 otherwise. |
| $\Delta_i$ | $\max\left\{\text{cpld}(a_1) - \text{cpld}(a_i), \frac{1}{K-1}\right\}$ |
| $H_i$ | $\sum_{j \neq i} \frac{1}{\Delta_{ij}^2}$ |
| $H_\infty$ | $\max_i H_i$ |
| $\Delta_i^\epsilon$ | $\max\{\Delta_i, \epsilon(1 - \text{cpld}(a_1))\}$ |