[Reviews · NeurIPS 2015]

Submitted by Assigned_Reviewer_1

Good paper on an interesting and up-to-date topic, albeit not so easy to read.
Summary: The authors address the duelling bandits problem without assuming the existence of a Condorcet winner. They propose new algorithms and derive regret bounds.

Submitted by Assigned_Reviewer_2

This paper tackles the dueling bandits problem, focusing on the scenarios where a Condorcet winner does not exist. In particular it focuses on finding the Copeland winner (i.e., the arm that beats maximal number of other arms) and proposes two algorithms for this task. The paper provides theoretical bounds on these algorithms which indicate better asymptotic regret performance than previous approaches for identifying the Copeland winner.

Overall the paper (with the supplementary material) was quite solid in terms of material. However I did have some significant concerns.

I wanted to start my critique with the informativeness and "quality" of the actual 9 page material. I recognize that this is a matter of principle to which the authors may object, but in my honest opinion I would like to reward the authors and NIPS submissions that took the extra effort of making a paper self-contained and informative within the 9 page limit, although that may have meant leaving out certain interesting results/analysis/experiments. At the end of the day what gets published is the 9 pager, which is what I believe should be the determining factor of a paper's quality. After all the NIPS guidelines clearly state that I reserve the right to do so:

"Note that the reviewers and the program committee reserve the right to judge the paper solely on the basis of the 9 pages of the paper; looking at any extra material is up to the discretion of the reviewers and is not required."

With this in mind, I actually find the overall quality and clarity of the 9 pager itself fairly lacking. Most of the interesting analysis and all the other empirical evidence are hidden away in the 20+ page appendix. In fact the overall analysis of the algorithms itself is fairly hard to understand as the 9-pager itself presents it in a rather hard-to-understand manner. Most of the paper constantly makes forward references to the supplementary material to complete the story, though its' purpose is to "supplement" the main content not serve as a continuation of it.

Some of the things which I would have liked to see in the actual 9-pager itself would include:

a) Significance: Why is this work significant? Lately the work on dueling bandits is becoming increasingly incremental with different approaches improving marginally over the previous on different metrics. So why is the Copeland winner what we should really be interested in as compared to the Borda winner or the von Neumann winner or some yet-to-be-published winner? Because at the end of the day what a practitioner cares about is the method that works best in practice not the specific metric that was used to get to that method. Towards this, I don't really find this paper providing any compelling evidence that the Copeland winner is the right one to aim for.

b) (Quality+Significance) Related to the above, but why would Condorcet methods not work well in cases such as this (even though the Condorcet winner may not exist). The authors mention that these methods may fail for these problems, but the 9 pager itself does not contain any evidence (theoretical or empirical) indicating the same.

c) (Quality+Clarity) Providing empirical evidence in the main material (even a small study whose setup can be detailed in the appendix) could convince a practitioner to switch such a method. A set of involved hard-to-gauge theoretical results alone do not constitute a compelling enough set of evidence in my opinion.

Again while I understand that the authors have addressed some of these criticisms in the supplementary material, I am basing this review on the 9 page material to be fair to all the other papers I review which have relied on me reading just the 9 pages and not a 32 pager.

Summary: This paper proposes theoretically sound solutions to well-established problem (dueling bandits). While the methods and accompanying analysis (in the appendix) are sound, what concerns me here is the fact that the actual 9-page content of the paper is not informative enough and instead relies on a 20+ page appendix to make the paper self-contained.

Submitted by Assigned_Reviewer_3

The paper considers the dueling bandit problem where the Condorcet winner does not exist, but instead the Copeland winners are considered to be optimal arms. The authors come up with two algorithms called CCB and SCB whose regret bounds are of order K^2 + K log K and K log K log T respectively which is a striking result in the case of Copeland ranking.

The CCB algorithm is based on the mixture of the principle of ``optimism followed by pessimism''. More detailed, it first selects an arm to be compared based on the optimistic estimate of the preference matrix. That is, it selects one of the possible Copeland winners. The other arm to be compared is chosen based on the pessimistic estimate of preference matrix, that is, it selects an arm which likely rejects the hypothesis of that the first arm that is already chosen is indeed a Copeland winner.

The SCB algorithm is an explore and then exploit algorithm. The core of the proposed method is a PAC algorithm which identifies a Copeland winner with high probability. This PAC algorithm is repeatedly called with a exponentially increasing budget.

The paper is very dense. I have to admit that I did not check the whole appendix which consists of 20 pages!! I read through the experimental study which is enjoy to read, but unfortunately the authors had decided to defer it to the supplementary material. The analysis is very rigorous the main steps of the proof can be understood without the appendix. I liked the paper in general thus I have only some minor comments.

Comments:

In the pseudo-code of CCB, I was not able to find out the role of line 9A. Why it is needed to reset these sets? Please make this issue clear in the text.

Appendix C.2, Table 1: % signe should be put after the number in the cell which is concerning to Random Walk - Borda
Summary: The paper is very dense but at the same time it is well-written and technically sound.

Submitted by Assigned_Reviewer_4

This paper proposes two algorithms for the copeland dueling bandits problem.

The two algorithms are analyzed theoretically and some experimental results are contained in the supplementary material.

From a theoretical perspective, this paper contains some nice work. The algorithms are intuitive, although I'm surprised SCB works so well.

Perhaps the authors can discuss some intuition in more depth?

My main critiques come from the writing style.

Most notably, the second half of the main paper is not very readable, with statement of complicated lemma after complicated lemma.

I think the authors should strongly consider moving these all to the appendix, and replacing them with much coarser statements that provide more intuition.

This paper is generally lacking in providing intuition.

The description of Algorithm 2 is a bit weird, since it calls an algorithm in the appendix.

I don't really see the necessity of this.

I think the authors should find space to move some of the empirical results to the main body of the paper.

Perhaps also smaller/simplified versions of Figure 7 & 8.

More broadly, the comparison between Copeland winner and von Neumann winner is interesting.

It's quite surprising that they match so often in your experiments, since I believe (correct me if I'm wrong) the von Neumann winner is often not deterministic.

Perhaps the authors could comment on this further.

Citation 19 needs to be updated.

*** RESPONSE TO AUTHOR REBUTTAL *** I thank the authors for their rebuttal.

I strongly encourage the authors to restructure their paper to be more clear.
Summary: The paper gives a solid theoretical result.

I think the paper could be better written.

Author Feedback
Author rebuttal: General comments:

We would like to thank the reviewers for their insightful comments. We're pleased that the technical contribution was so well received. Regarding the issue of balancing content in the main paper vs. the appendix, we'd like to point out that it is straightforward for us to adjust this in the CRC. In particular, we could easily:
1- Move most of the technical lemmas to the appendix and replace them with more intuitive explanations of the results.
2- Move some of the experimental results to the main paper.
3- Include in the main paper a compare and contrast between the various dueling bandit solution concepts.

Specific comments:

Reviewer 1:

a) The goal of the paper is not to show that the Copeland winner is a superior solution concept. On the contrary, many solution concepts have good arguments in their favor, and the trade-offs between them is a subtle subject that has occupied the field of social choice theory for decades.

That said, there is a clear argument in favor of the Copeland winner over the Condorcet winner, namely that it doesn't require assuming a Condorcet winner exists. Since this assumption was the main barrier to the practical application of Condorcet methods, and we provide the first efficient Copeland method, thereby removing this barrier, we feel that the contribution of the paper is not incremental at all.

We hope our "general comments" above address your comments (b) and (c), as the experimental results clearly demonstrate the hazard of using algorithms such as RUCB that rely on the Condorcet assumption.

Reviewer 2:
Line 9A in Alg 1 is a cautionary measure that resets the sets B^i if at some point it turned out that one of the arms in B^i beats arm a_i by a wide margin instead of losing to it, which is what the arms in B^i are supposed to do. We will clarify this in the CRC. Also, thanks for pointing out the typo.

We hope our "general comments" above address your other concerns.

Reviewer 3:
Regarding the Copeland winner vs. the von Neumann winner, note that the numbers in the second row of Table 1 in Appendix C.2 measure when the support of the von Neumann winner intersects with the set of winners according to each definition. Moreover, the following paragraph states that in 94.2% of cases, the Copeland winners were completely contained inside the support of the von Neumann winner.

We hope our "general comments" above address your other concerns.

Reviewer 4:
We are unsure how to react to this review. In the absence of specific examples of problems with the writing, we cannot respond meaningfully. However, the other reviewers did not mention any problems with sentence construction. Furthermore, the paper was written and edited by native English speakers. While there may be a few inevitable typos, in general we are confident the paper is grammatically correct.

Moreover, regarding the claim that the results can be obtained "using the usual confidence bound approach for preference learning with a slight modification", note that the direct confidence bound approach leads to algorithms with regret O(K^2 log T). As pointed out repeatedly in the paper, including in the abstract, the main novelty of the algorithms and the analyses presented here is the fact that they have a linear dependence on K, and this is achieved using ideas that are orthogonal to the standard confidence bound approach. Indeed, people familiar with the problem often find it surprising that a linear dependence in K is even possible, so we believe that describing our results as "slight modifications" is simply not justified, and urge Reviewer 4 to discuss the matter with the other reviewers and reconsider.

Reviewers 5 and 6:
Thank you for your comments. Please let us know if you have any recommendations for the improvement of the exposition that are not addressed by the "general comments" above.